# Hepatocellular senescence induces multi-organ senescence and dysfunction via TGFβ

Christos Kiourtis [1,2], Maria Terradas-Terradas [1,2], Lucy M. Gee[3], Stephanie May [1,2], Anastasia Georgakopoulou [1], Amy L. Collins [3], Eoin D. O'Sullivan [4,5], David P. Baird[4], Mohsin Hassan[6], Robin Shaw[1], Ee Hong Tan[1], Miryam Müller [1], Cornelius Engelmann[6], Fausto Andreola[7], Ya-Ching Hsieh[2], Lee H. Reed[3,8], Lee A. Borthwick[3,8], Colin Nixon [1], William Clark[1], Peter S. Hanson[9], David Sumpton [1], Gillian Mackay[1], Toshiyasu Suzuki [1], Arafath K. Najumudeen [1], Gareth J. Inman [1,2], Andrew Campbell [1], Simon T. Barry [10], Alberto Quaglia[11,12], Christopher M. Morris [13], Fiona E. N. LeBeau[13], Owen J. Sansom [1,2], Kristina Kirschner [1,2,14,15,16], Rajiv Jalan [7,17], Fiona Oakley [3] & Thomas G. Bird [1,2,4] ✉

Cellular senescence is not only associated with ageing but also impacts physiological and pathological processes, such as embryonic development and wound healing. Factors secreted by senescent cells affect their microenvironment and can induce spreading of senescence locally. Acute severe liver disease is associated with hepatocyte senescence and frequently progresses to multi-organ failure. Why the latter occurs is poorly understood. Here we demonstrate senescence development in extrahepatic organs and associated organ dysfunction in response to liver senescence using liver injury models and genetic models of hepatocyte-specific senescence. In patients with severe acute liver failure, we show that the extent of hepatocellular senescence predicts disease outcome, the need for liver transplantation and the occurrence of extrahepatic organ failure. We identify the TGFβ pathway as a critical mediator of systemic spread of senescence and demonstrate that TGFβ inhibition in vivo blocks senescence transmission to other organs, preventing liver senescence induced renal dysfunction. Our results highlight the systemic consequences of organ-specific senescence, which, independent of ageing, contributes to multi-organ dysfunction.

Cellular senescence is a state of permanent cell cycle arrest accompanied by a hyper-secretory phenotype (senescence-associated secretory phenotype, SASP) and is associated with both injury and ageing-related pathologies within affected organs[1,2]. Removal of senescent cells is beneficial to both organ function and organism survival[3–6]. Severe acute injury of any large organ is associated with systemic effects including multi-organ failure, of which acute liver failure (ALF) is a paradigm. ALF is itself associated with senescence induction and subsequent regenerative failure[7]. Studies, both in vitro and in vivo, have shown that senescence can be transmitted in a paracrine manner within affected organs[8–11]; however, whether senescence can spread systemically to distant organs remains unknown. Here, we use acute liver senescence as an exemplar model, independent of systemic ageing, to test whether senescence can be transmitted between organs in an endocrine manner. We demonstrate that systemic transmission of senescence affects multiple organs associated with target organ dysfunction and identify

**Fig. 1 | Hepatocellular senescence results in senescence and dysfunction in other organs. a**, A schematic of the experimental approach; 8–12-week-old $Mdm2^{E5/E6fl}$;$R26^{LSL-tdTomato/LSL-tdTomato}$ mice were intravenously injected with $2 \times 10^{11}$ GC of AAV-Cre or AAV-Null and culled 4 days later. The downstream targets of MDM2 are highlighted. **b**, Representative images of p21 IHC in liver cells of ΔMdm2^Hep and control mice; $n = 16/19$ control/ΔMdm2^Hep mice, respectively. **c**, Automated quantification of p21$^+$ liver cells; $n = 4/5$ control/ΔMdm2^Hep mice, respectively; unpaired two-tailed $t$-test. **d**, Representative images of p21 IHC on mouse kidney sections. **e**, Manual quantification of p21$^+$ renal tubular cells; the data are presented per field of view (FOV), $n = 16/19$ control/ΔMdm2^Hep mice (additional controls shown in Extended Data Fig. 1g); Welch's two-tailed $t$-test. **f**, Representative images of p21 IHC on brain sections. **g**, Manual quantification of p21$^+$ brain cells; $n = 4/6$ control/ΔMdm2^Hep mice, respectively; two-tailed Welch's $t$-test. **h**, Representative images of p21 IHC in lung sections. **i**, Automated quantification of p21$^+$ lung cells; $n = 6/9$ control/ΔMdm2^Hep mice, respectively; unpaired two-tailed $t$-test. **j**, Representative images of p21 IHC in liver sections of Kras^G12D/Kras^WT mice. **k**, Automated

quantification of p21$^+$ liver cells; $n = 8/11$ Kras^WT/Kras^G12D mice, respectively; two-tailed Mann–Whitney test. **l**, Representative images of p21 IHC on kidney sections of $Kras^{G12D}$/$Kras^{WT}$ mice. **m**, Manual quantification of p21$^+$ renal tubular cells; $n = 14/16$ Kras^WT and Kras^G12D mice, respectively; two-tailed Mann–Whitney test. **n**, The plasma levels of cystatin C in ΔMdm2^Hep or control mice 4 days post AAV-Cre or AAV-Null injection; $n = 9/10$ control/ΔMdm2^Hep mice, respectively; two-tailed Welch's $t$-test. **o**, The urine levels of alanine and threonine in ΔMdm2^Hep mice pre and post AAV-Cre injection; the dots represent the average peak area of $n = 3/4/5$ mice at days −2 and 0, 4 and −1 and 3, respectively; two-way ANOVA comparing each timepoint to induction day (day 0). **p**, The proportion of time spent by the ΔMdm2^Hep/control mice in the new arm of the Y maze at day 4; $n = 7/9$ control/ΔMdm2^Hep mice, respectively; unpaired two-tailed $t$-test. **q**, The share of stable versus unstable oscillations of hippocampal brain slices; $n = 13/14$ brain slices from four control/ΔMdm2^Hep mice each, respectively. All the bars are mean ± s.e.m., each dot represents one sample per mouse and the numbers are $P$ values. The scale bars are 50 μm and 5 μm in the inset magnifications. The source numerical data are available in Source data.

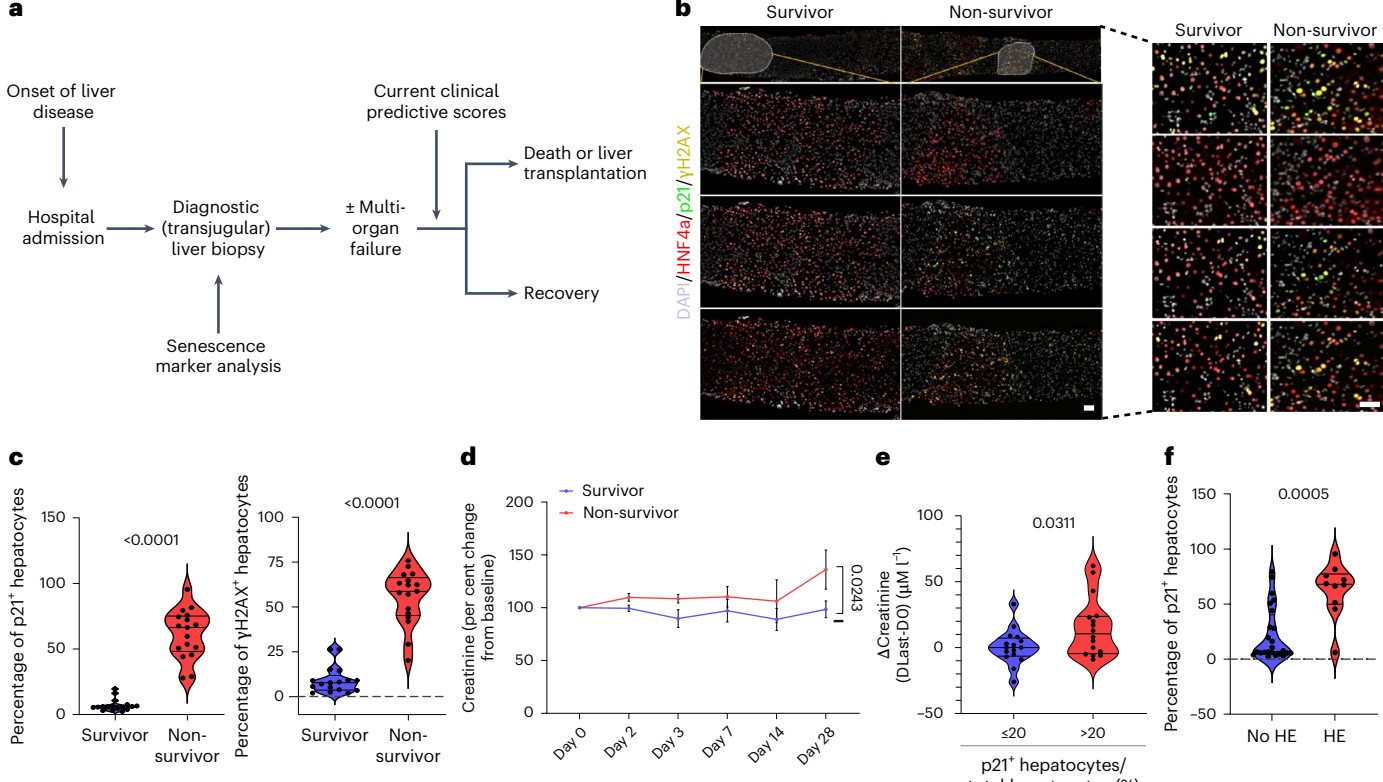

**Fig. 2 | Hepatocellular senescence in human severe acute indeterminate hepatitis correlates with development of subsequent renal dysfunction.** **a**, A schematic of patient stratification and outcomes. **b**, Multiplex immunofluorescence staining for hepatocytes (HNF4α, red), senescent cells (p21, green), DNA damage (γH2AX, yellow) and nuclei (DAPI, grey) in human patient liver tissue. **c**, Quantification of p21⁺ and γH2AX⁺ hepatocytes in survivors versus non-survivors. $n = 17$ patients in both groups; two-tailed Welch's $t$-test. **d**, The serum creatinine levels in survivors versus non-survivors over the course of 28 days post hospital admission. $n = 16/14$, $13/13$, $13/14$, $12/12$, $10/6$ and $9/5$

patients on days 0, 2, 3, 7, 14 and 28 for the survivors/non-survivors groups, respectively; two-tailed paired $t$-test. The bars are the mean ± s.e.m. **e**, Change in serum creatinine from first (D0) to last recorded (DLast) day of admission; two-tailed paired $t$-test. $n = 16$ in both groups. **f**, Quantification of p21⁺ hepatocytes in patients who developed hepatic encephalopathy (HE) versus those ones who did not. $N = 10$ and $n = 24$ in the 'HE' and 'no HE' groups, respectively; two-tailed Mann–Whitney test. On all violin graphs, each dot represents one biological sample from individual patients, and the numbers are $P$ values. Scale bars, 50 μm. The source numerical data are available in Source data.

the transforming growth factor β (TGFβ) signalling pathway as a critical mediator of this process.

## Liver senescence spreads to other organs

To model tissue-restricted senescence we used a genetic mouse model of conditional, hepatocyte-specific excision of the p53-binding domain of MDM2 ($Mdm2^{E5/E6fl}$; $R26^{LSL-tdTomato/LSL-tdTomato}$ mice) (Fig. 1a). Intravenous administration of the hepatocyte-specific AAV8-TBG-Cre vector resulted in widespread genetic recombination in the hepatocytes (ΔMdm2^Hep), unlike the empty vector AAV8-TBG-Null (control) (Extended Data Fig. 1a,b). The genetic recombination was hepatocellular specific, with no detectable Cre-mediated recombination occurring in extrahepatic tissues, as previously reported[12] (Extended Data Fig. 1b–f). MDM2 inactivation resulted in p53 accumulation in the hepatocytes and subsequent upregulation of its target gene, the cell cycle inhibitor p21, which is itself a key marker of senescence (Fig. 1b,c and Extended Data Fig. 1a,b). This established a senescent phenotype in the liver manifested by increased senescence-associated β-galactosidase (SA β-Gal) activity and a senescence-associated transcriptome (Extended Data Fig. 2a–c).

We next examined other organs for senescence induction in this model. In the kidney, we observed an increase in p21 expression and in SA β-Gal activity, along with a senescent transcriptional signature (Fig. 1d,e and Extended Data Figs. 1g and 2d,e). Additionally, kidneys of ΔMdm2^Hep mice contained increased concentration of several SASP markers (Extended Data Fig. 2f). In addition to the kidney, we observed increased expression of p21 in the brain and the lungs of ΔMdm2^Hep mice

(Fig. 1f–i). Therefore, in this model, we observe renal, brain and lung senescence in response to acute genetically induced and hepatocellular restricted senescence.

In ΔMdm2^Hep mice, p53 accumulation in hepatocytes also induces moderate liver injury and dysfunction as evidenced by an increase in plasma levels of alanine aminotransferase (ALT), alkaline phosphatase (ALP) and bilirubin, as well as an increase in hepatic cleaved caspase 3 (CC3), a marker of apoptosis (Extended Data Fig. 2g–j). We also examined a further liver injury and senescence model induced by carbon tetrachloride (CCl₄)[7], observing here that renal p21 expression and dysfunction also occurred (Extended Data Fig. 2k–n). To address whether systemic transmission of senescence is driven by liver senescence or by liver injury, we induced liver senescence in the absence of injury through oncogene activation. Using hepatocyte-specific expression of oncogenic $Kras^{G12D}$, we observed liver senescence without histological or biochemical evidence of liver injury or dysfunction (Fig. 1j,k and Extended Data Fig. 2o–r). Importantly, similar to the ΔMdm2^Hep model, renal p21 expression was increased in mice that expressed KRAS^G12D in their hepatocytes ($Kras^{G12D}$ mice) compared with the control ($Kras^{WT}$) mice (Fig. 1l,m and Extended Data Fig. 2s,t).

When near-global hepatocellular recombination (~80%) of hepatic Mdm2 occurs[13], the animals become systemically unwell and require to be killed humanely 4 days following induction. To observe the effects of hepatic senescence in a longer-term model, we titrated the hepatic induction through reduction of the AAV8-TBG-Cre vector. When approximately 67% of hepatocytes are induced (so-called

recovery-Mdm2 model), a reduction in p21+ hepatocytes was observed at day 4 compared with the ΔMdm2Hep model (Extended Data Fig. 3a–c). In the recovery-Mdm2 model, we tracked renal p21 expression over time, observing a delayed p21 induction, again associated with hepatic injury and senescence. This resolved as the hepatic senescence signature was lost over time (Extended Data Fig. 3b–e). When p53-driven hepatic senescence was further titrated down, the renal signature was lost (Extended Data Fig. 3d), as it was in the KrasG12D model also. Therefore, we conclude that a critical mass of liver senescence is able to drive the systemic spread of senescence to multiple distant organs.

## Transmitted senescence is associated with organ dysfunction

We then proceeded to explore whether extrahepatic senescence was associated with organ dysfunction. To do this, we first measured plasma cystatin C, a marker of renal filtration efficiency, and the levels of urine amino acids as a readout for renal tubular function. Plasma cystatin C and several urinary amino acids were perturbed in the ΔMdm2Hep mice, consistent with renal dysfunction (Fig. 1n,o and Extended Data Fig. 4a). Urinary amino acid loss was also observed in the recovery-Mdm2 model, temporally consistent with the resolution of the renal senescent phenotype (Extended Data Fig. 3f). To assess brain functionality, we performed cognitive function studies. First, we utilized the Y-maze test, which assesses the natural exploratory instinct of mice. Normal murine behaviour consists of preferential exploration of the novel arm of a Y maze after an acclimatization period before opening the novel arm. ΔMdm2Hep mice spent less time in the novel arm compared with control mice, consistent with impaired cognitive function (Fig. 1p and Extended Data Fig. 4b). To further corroborate this finding, we performed electrophysiology on hippocampal slices from these mice and measured the gamma (30–80 Hz) oscillation area power and frequency in response to a cholinergic agonist, carbachol. Following carbachol administration, brain sections from healthy mice (control) produced network gamma oscillations of increasing power and stable frequency before stabilizing after 120–150 min, as expected (Extended Data Fig. 4c,d). In comparison, brain sections from ΔMdm2Hep mice exhibited weaker and less stable oscillations consistent with abnormal hippocampal function (Fig. 1q and Extended Data Fig. 4c,d). Hence, the senescence observed in multiple organs in response to liver senescence is associated with organ-specific dysfunction.

To explore whether these findings are of functional relevance in human disease, we interrogated a cohort of patients with acute liver-specific disease (acute indeterminant hepatitis), each undergoing diagnostic liver biopsy in the early stages of this severe disease (Fig. 2a). Consistent with previous reports, routine biochemistry and clinical parameters at clinical presentation did not define either outcome or subsequent multi-organ dysfunction (including both renal and cerebral failure—hepatic encephalopathy) in this cohort (Supplementary Table 1). We find, however, that elevated levels of hepatocellular senescence markers p21 and γH2AX on the diagnostic biopsy predict subsequent survival (Fig. 2b,c). Serum creatinine levels, indicative of

renal dysfunction, increased over time in patients who did not survive (Fig. 2d), and liver biopsy p21 levels in hepatocytes correlate with both subsequent renal (Fig. 2e) and cerebral dysfunction (Fig. 2f). In contrast to prior clinical scoring systems, which use markers during progressive disease in its later stage to define patient outcome and the need for transplantation, hepatic p21 levels in this cohort predict this on admission with features of severe acute hepatitis.

## SASP factors induce the systemic transmission of senescence

To uncover the changes occurring to the kidney in response to liver senescence, we performed single-cell RNA sequencing (scRNA-seq) on kidneys of ΔMdm2Hep and control mice (Fig. 3a). From a total of 30,236 single-cell transcriptomes and after stringent quality control (Extended Data Fig. 5a), 24,215 cells were used for downstream analysis. Initial clustering of the control and ΔMdm2Hep cells together resulted in eight clusters (Extended Data Fig. 5b–e), with each cluster containing cells both from ΔMdm2Hep and from control mice. Again, there was no evidence of activity of the AAV8-TBG-Cre vector on the genetic reporter in the kidney (Extended Data Fig. 5f).

By colouring the cells on the uniform manifold approximation and projection (UMAP) plot depending on experimental cohort (control or ΔMdm2Hep), we observed that, particularly in the broad cluster of proximal tubular cells (PTC), the ΔMdm2Hep cells deviated from the control cells indicating a shift in their transcriptome (Fig. 3b). To further investigate this, we performed unsupervised pathway analysis in the PTC cluster, comparing ΔMdm2Hep and control cells. This revealed several pathways associated with senescence (wound healing response and down-regulation of apoptosis) were enriched in the ΔMdm2Hep PTCs (Extended Data Fig. 6a). Next, to identify the cell identity of the p21-expressing cells we observed in kidney tissue by immunohistochemistry (IHC), we created a p21 gene signature comprised of genes associated with p21 expression. Most cells with a high score for this signature (p21high cells) belong to the PTC compartment, in agreement with the p21 IHC in kidney sections that showed preferential localization of the p21+ cells in the renal tubules (Fig. 3c and Extended Data Fig. 6b). In addition, there was a significant increase of p21high cells in the ΔMdm2Hep PTCs identified in the scRNA-seq data (Fig. 3d), consistent with the increase in p21+ kidney cells observed in tissue. Additionally, using a second transcriptional senescence signature specifically of renal tubular senescence, developed from a murine renal injury induced senescence and validated in both murine renal injury and ageing and senescent human renal epithelium[14], we also observed an increase in senescent epithelial cells in the kidney following hepatic senescence (Fig. 3e and Extended Data Fig. 6c) but identify these as a generally separate population to those PTC with the p21 gene signature (Extended Data Fig. 6d).

An important function of PTCs is the uptake of filtered substances, including amino acids. As we observed suppression of pathways related to polarity and amino acid transport in the ΔMdm2Hep PTCs (Extended Data Fig. 6a), we examined the expression of several transporters of the solute carrier (Slc) family across the different parts of the renal tubule.

**Fig. 3 | scRNA-seq reveals transcriptional changes, including amino acid transporter expression within the proximal tubular compartment in response to liver senescence. a**, A schematic of the scRNA-seq experiment. Three kidneys from control mice and three kidneys from ΔMdm2Hep mice were dissociated, and droplet-based scRNA-seq was performed on them on a 10x Chromium chip. **b**, UMAP plots of all 24,215 cells (control and ΔMdm2Hep). The cells are coloured on the basis of the broad clusters (left) or experimental group (right) (green, control; blue, ΔMdm2Hep). **c**, UMAP plots showing the distribution of the cells that have a positive score for the p21 gene signature in the control (top) and the ΔMdm2Hep (bottom) cells. The inset image is a representative kidney section from a ΔMdm2Hep mouse stained for p21 by IHC with highlighted tubules (dashed lines). **d**, Pie charts showing the share of p21high, TGFβhigh PTCs and JAK–STAThigh PTCs and mesenchymal cells in the control and ΔMdm2Hep

samples. Contingency was tested with the chi-square test (one-tailed). **e**, A bar chart showing the total number of cells with a positive score for the senescence transcriptional signature in the control and ΔMdm2Hep samples (chi-square test (one-tailed)). **f**, A heat map showing the significantly differentially expressed Slc transporter genes in the PTC, DTC and LOH compartments. The gene IDs in the red frames are genes encoding transporters associated with amino acid transport. **g**, A schematic of the p21KO experiment. Six Mdm2E5/E6fl; p21WT and six Mdm2E5/E6fl; p21KO mice were injected with AAV-Cre and culled 4 days later. **h**, The plasma levels of cystatin C in Mdm2E5/E6fl; p21WT and Mdm2E5/E6fl; p21KO mice 4 days post AAV injection. n = 6 for both groups; unpaired two-tailed t-test. The bars are the mean ± s.e.m., and the numbers on the graphs are P values. The source numerical data are available in Source data.

We observed marked differential expression of several genes encoding SLC transporters specifically in the PTCs but not in the distal tubular cells (DTCs) or in the loop of Henle (LOH) (Fig. 3f). Some of these transporters (namely, SLC6A6, SLC7A7, SLC25A39, SLC7A8 and SLC7A13) are involved in amino acid transport in the PTCs[15–17], consistent with the increase in amino acids observed in the urine of ΔMdm2^Hep mice. We also observed that in the p21^high PTCs, the expression of several *Slc* transporter genes was altered compared with PTCs without the p21 signature and

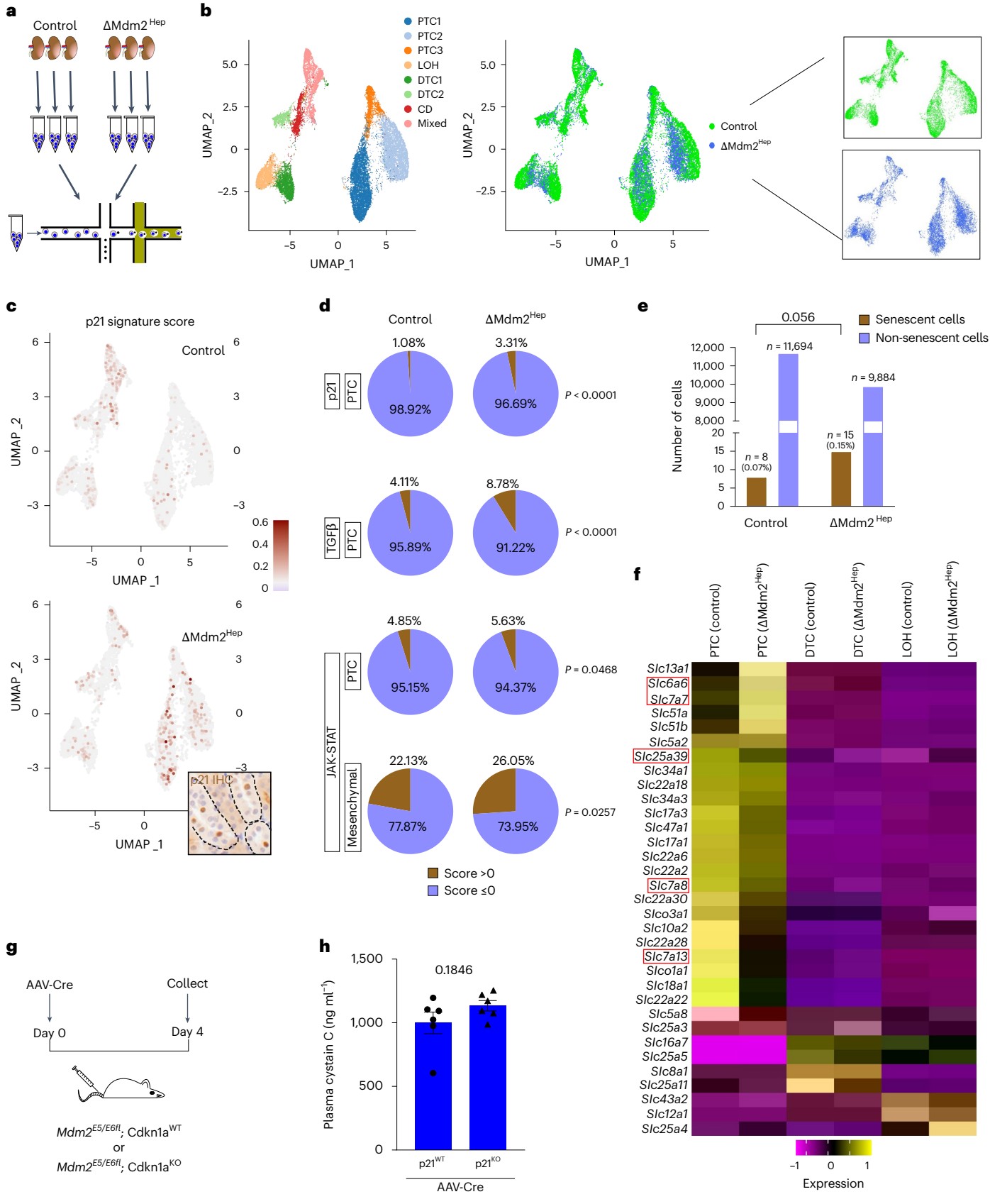

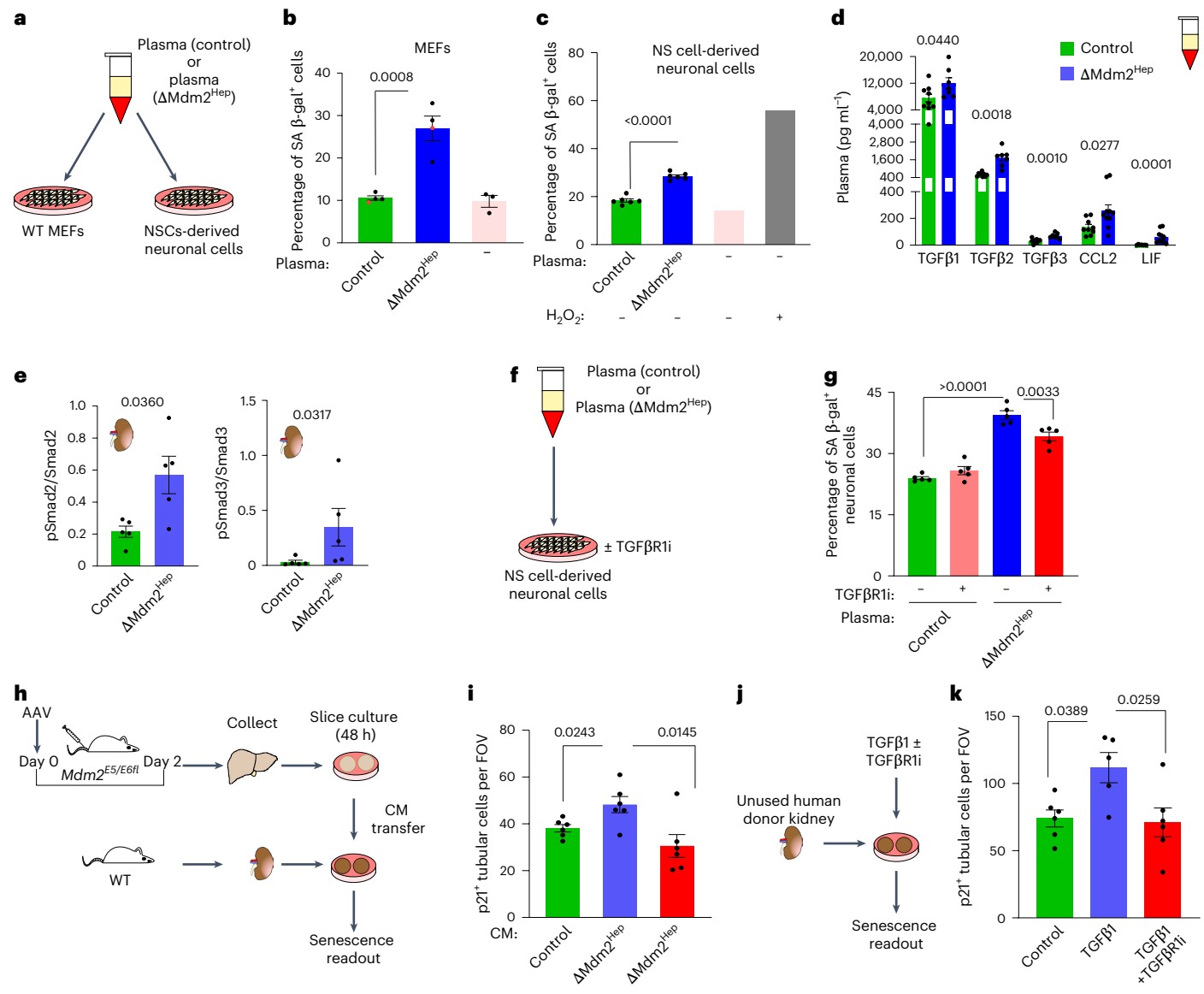

**Fig. 4 | Hepatic SASP in plasma induces senescence. a**, A schematic of the in vitro plasma treatment. WT MEFs and neuronal cells were treated with plasma from control/ΔMdm2^Hep mice and stained for SA β-Gal. **b**, Manual quantification of SA β-Gal⁺ WT MEFs. Each dot represents one technical replicate. Each of the two biological replicates (one mouse's plasma sample, with 1/3 technical replicates shown as red/black points, respectively) from both control/ΔMdm2^Hep; *n* = 3 technical replicates for no plasma control. One-way ANOVA compares all the technical replicates. **c**, Manual quantification of SA β-Gal⁺ NS cell-derived neuronal cells. Each dot represents the average of the technical triplicates of one biological replicate for each group (*n* = 6 biological replicates). For the 'no plasma' groups, each bar is the mean of four technical replicates (*n* = 1 biological replicate), one-way ANOVA. **d**, The plasma levels of TGFβ1, TGFβ2, TGFβ3, CCL2 and LIF. For the TGFβ ligands: *n* = 9/7, CCL2: *n* = 9/10, LIF: *n* = 7/10 control/ΔMdm2^Hep plasma samples, respectively; unpaired two-tailed *t*-test, Welch's *t*-test and Mann–Whitney *U* test for TGFβ1, TGFβc3 and TGFβ2 and CCL2 and LIF, respectively. Each dot represents one biological replicate (one mouse's plasma sample). **e**, Quantification of western blot band signal intensity (Extended

Data Fig. 8c) normalized to β-actin; *n* = 5 each, two-tailed Welch's *t*-test/Mann–Whitney for pSMAD2/SMAD2/pSMAD3/SMAD3. **f**, A schematic of TGFβ inhibitor treatment. **g**, Plasma-treated neuronal cells were treated with either TGFβR1i (AZ12601011, AstraZeneca) or vehicle (DMSO), stained for SA β-Gal and manually quantified. Each dot represents the average of a technical triplicate from one biological replicate (one mouse's plasma sample); n = 5 each, one-way ANOVA. **h**, A schematic of the transfer of conditioned media (CM) in murine slice culture experiments. The livers of *Mdm2*^E5/E6fl/*R26*^LSL-tdTomato/LSL-tdTomato mice were collected 2 days after AAV-Null/AAV-Cre (*n* = 6 each). CM was collected 48 h later and added, with TGFβR1i or vehicle, to ex vivo-cultured WT murine kidney slices. **i**, Manual quantification of kidney slice p21⁺ cells, *n* = 6 each; unpaired two-tailed *t*-tests. **j**, A schematic of a human kidney slice experiment. **k**, Human kidney slices were treated with TGFβ1 with or without TGFβR1i ex vivo, and p21⁺ renal cells were manually quantified; *n* = 6 each group, one-way ANOVA. All the bars are the mean ± s.e.m., and the numbers on the graphs are the *P* values. The source numerical data are available in Source data.

specifically in the ΔMdm2^Hep mice (Extended Data Fig. 6e). Therefore, the effects of liver senescence upon the kidney are particularly focused upon the PTCs, resulting in impaired PTC polarity and function.

Next, to explore the p21 dependence of renal senescence and dysfunction, we performed the ΔMdm2^Hep model on a p21-knockout

(p21^KO) background (*Mdm2*^E5/E6fl; *Cdkn1a*^KO). Here, we observed a lack of reversal in the renal dysfunction observed in ΔMdm2^Hep animals (Fig. 3g,h) implying that secondary organ dysfunction is not dependent upon p21 in either the primary senescent (hepatocyte) or the secondary organs.

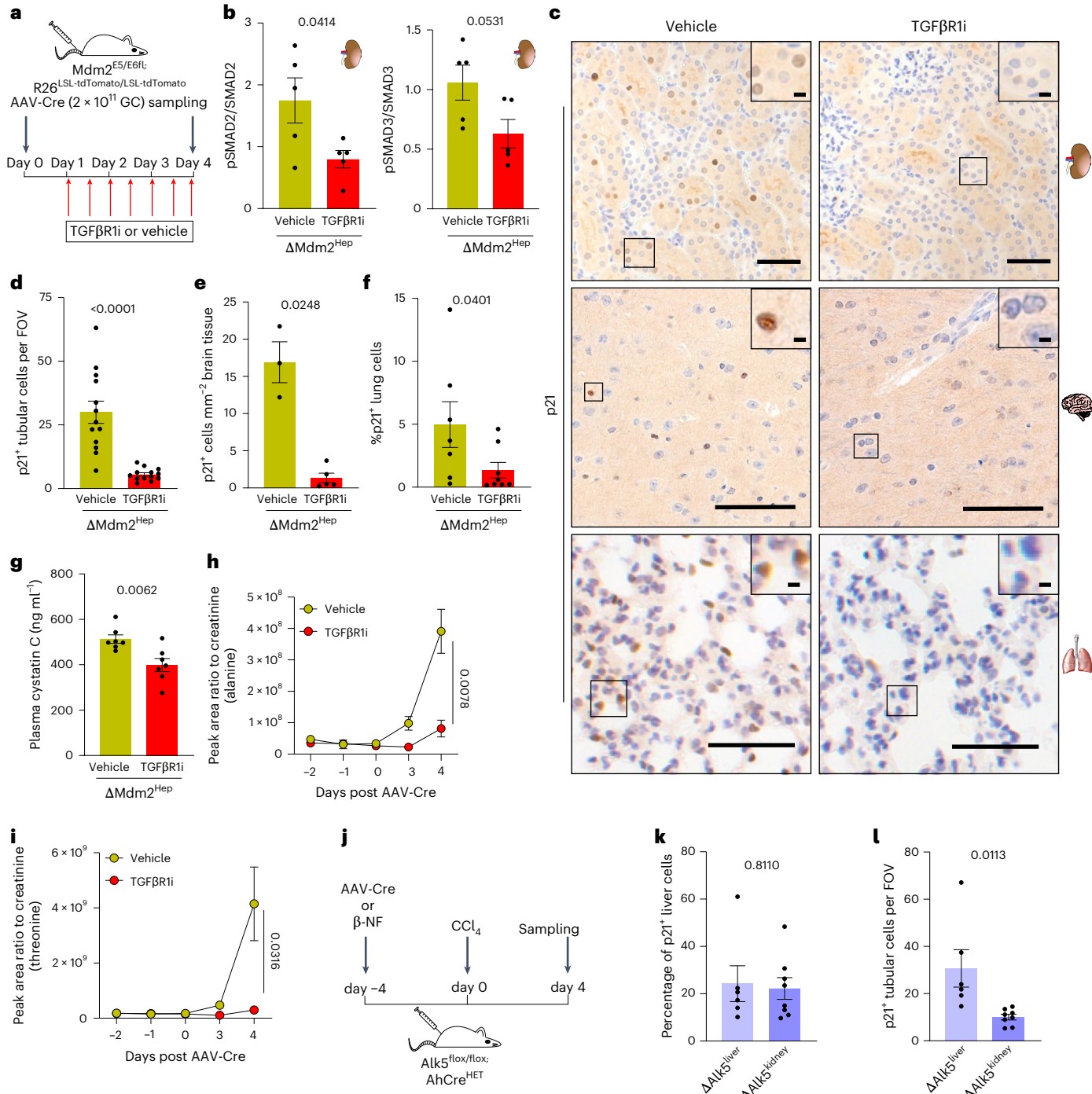

**Fig. 5 | Inhibition of TGFβ signalling prevents induction of senescence in the extrahepatic organs. a**, A schematic of the in vivo treatment with TGFβ receptor inhibitor (TGFβR1i). ΔMdm2^Hep mice were treated with TGFβR1i or vehicle by oral gavage twice daily, starting 24 h post AAV-Cre. **b**, A quantification of western blot band signal intensity (Extended Data Fig. 8a); $n = 5$ each group, unpaired two-tailed $t$-tests. **c**, Representative images of p21 IHC in kidney, brain and lung sections of ΔMdm2^Hep mice treated with vehicle or TGFβR1i ($n = 3$–13 mice for each group as stated below). **d**, A manual quantification of p21⁺ renal cortical cells; the data are presented as mean p21⁺ cells per field of view (FOV) for each mouse; $n = 13$ (biological replicates) mice for each group, two-tailed Mann–Whitney test. **e**, a Manual quantification of p21⁺ brain cells; the data are presented as p21⁺ cells per square millimetre, $n = 3/5$ vehicle/TGFβR1i treated, respectively; two-tailed Welch's $t$-test. **f**, Automated quantification of p21⁺ lung cells; $n = 7/8$ vehicle/TGFβR1i treated, respectively; two-tailed Mann–Whitney test. **g**, Plasma cystatin C of vehicle or TGFβR1i treated at cull. Each dot represents the data from one mouse; $n = 7$ mice for each group, unpaired two-tailed $t$-test.

**h,i**, The levels (peak area ratio to creatinine) of alanine (**h**) and threonine (**i**) in the urine of ΔMdm2^Hep mice before and after AAV-Cre injection; $n = 4/5$ vehicle/ TGFβR1i treated, respectively, per timepoint, two-tailed Welch's $t$-test comparing vehicle- or TGFβR1i-treated mice at day 4. **j**, A schematic of the genetic ablation of TGFβ signalling in the kidney. *TGFBR1(Alk5)^{fl/fl}* ± AhCre^{+/−} mice were administered either $2 \times 10^{11}$ GC ml⁻¹AAV-Cre (ΔAlk5^{Liver}) or $3 \times 80$ mg kg⁻¹β-NF (ΔAlk5^{Kidney}), respectively, or control induction ($2 \times 10^{11}$ GC ml⁻¹AAV-TBG-Null/vehicle, respectively) and were collected 4 days later. **k,l**, Automated quantification of p21⁺ liver cells (**k**) and manual quantification of p21⁺ renal cortical cells (**l**) in ΔAlk5^{Liver} and ΔAlk5^{Kidney} mice; the data are presented as percentage of total liver cells or as p21⁺ cells per FOV. $n = 6/8$ for ΔAlk5^{Liver}/ΔAlk5^{Kidney} mice, respectively; Brown–Forsythe and Welch ANOVA test (liver) or one-way ANOVA (kidney). The bars are the mean ± s.e.m., and the numbers on the graphs are $P$ values. The scale bars are 50 μm and 5 μm in the inset magnifications. The source numerical data are available in Source data.

To investigate the mechanism of senescence transmission from the liver to the kidney, we hypothesized that SASP factors secreted by the liver reach the kidney via the circulation. This hypothesis was supported by an enrichment of the ΔMdm2[Hep] liver transcriptome for a SASP gene signature (Extended Data Fig. 7a). To test this hypothesis, we treated cells in vitro with plasma from ΔMdm2[Hep] or control mice (Fig. 4a). We performed this using both wild-type (WT) murine embryonic fibroblasts (MEFs) and neuronal cells differentiated from human neural stem cells (NS cells). In both cases, we observed a notable increase in SA β-Gal activity after treatment with ΔMdm2[Hep] plasma compared with plasma from healthy controls (Fig. 4b,c). Next, we performed cytokine arrays on plasma from ΔMdm2[Hep] and control mice. While plasma concentration of many of the screened factors remained unchanged, we observed a significant increase in the levels of a number of classical SASP markers: TGFβ1, TGFβ2, TGFβ3, C–C motif chemokine ligand 2 (CCL2), hepatocyte growth factor, angiopoietin and leukaemia inhibitory factor (LIF) in the plasma of ΔMdm2[Hep] mice (Supplementary Table 2). We referred to our renal transcriptomics analysis to assess cross-correlation to the downstream pathways (that is, the TGFβ and the JAK–STAT (Janus kinase–signal transducer and activator of transcription) signalling pathways) stimulated by these factors in the kidneys of ΔMdm2[Hep] mice, observing elevations in both but, particularly, the TGFβ pathway in the relevant senescent PTC population (Extended Data Fig. 7b–e). The TGFβ pathway was also elevated at the whole kidney level in the transcriptomes of the ΔMdm2[Hep] (Extended Data Fig. 7f) and Kras[G12D] (Extended Data Fig. 2u) models and has previously been associated both with intra-organ paracrine senescence[7] and renal dysfunction[18].

We, therefore, further studied the mechanistic role of the TGFβ pathway in the induction of PTC senescence and dysfunction. We confirmed, by western blotting, that phosphorylated (p)Smad2 and pSmad3 levels (messengers of the canonical TGFβ pathway) were elevated on whole kidney lysates in the ΔMdm2[Hep] model (Fig. 4e and Extended Data Fig. 7g). At the single-cell level, we observed that the ΔMdm2[Hep] PTCs had a higher score for a transcriptional TGFβ gene signature compared with control PTCs (Fig. 3d) and this was validated in tissue with in situ hybridization for *Smad7*, a transcriptional target of TGFβ signalling being overexpressed in senescent PTCs (Extended Data Fig. 7b,c). The increase in plasma TGFβ ligands is associated with their increased hepatic production, as evidenced by their transcriptional upregulation in this organ (Extended Data Figs. 2c and 7h), including by a subset of senescent hepatocytes (Extended Data Fig. 7i) and, particularly, the non-parenchymal cells (Extended Data Fig. 7h). This also corresponded with increased expression of the TGFβ receptor (*Tgfbr1-3*), both in the liver and in the whole kidney, and by renal PTCs, specifically (Extended Data Fig. 7j–l).

We next performed functional studies using plasma and media transfer experiments to test the role of TGFβ for transmission of senescence from plasma to neurons and the kidney. Initially, we tested this in vitro in plasma-treated neuronal cells and observed a significant reduction in the number of SA β-Gal[+] cells after TGFβR1 inhibition (Fig. 4f,g). Next, using an ex vivo precision-cut liver and kidney slice (PCLS and PCKS) system, we tested if the circulating plasma changes came directly from the senescent liver (Fig. 4h). In this setting, transferring media from senescent PCLS to the PCKS induced renal senescence (Fig. 4i). We further tested TGFβ receptor inhibition, showing p21 induction was inhibited using the TGFβR1 inhibitor AZ12601011 (ref. 19). Next, for human relevance, we treated healthy human PCKS with the TGFβ ligand, inducing p21 expression, an effect also suppressed by TGFβR inhibition (Fig. 4j,k).

## TGFβ inhibition prevents systemic transmission of senescence

Having associated TGFβ pathway activation with senescence transmission in vitro, we went on to functionally test this pathway's role in senescence and dysfunction induction/transmission in vivo. Using the TGFβR1 inhibitor in the ΔMdm2[Hep] model (Fig. 5a) resulted in inhibition

of the TGFβ pathway in the kidney, as expected (Fig. 5b and Extended Data Fig. 8a) and a reduction p21[+] cells in the kidney, brain and lung of these mice (Fig. 5c–f) without affecting the genetically induced liver senescence (Extended Data Fig. 8b,c). A similar reduction in renal p21[+] cells was also observed in the Kras[G12D] model following TGFβR1 inhibitor (TGFβR1i) treatment (Extended Data Fig. 8d,e). The improvement of the senescence phenotype was accompanied by an improvement in renal function, as indicated by a significant reduction of plasma cystatin C and urine amino acids (Fig. 5g–i and Extended Data Fig. 8f). We tested the effects of senomorphics and senolytics in the ΔMdm2[Hep] model. Using rapamycin as a senomorphic did not affect genetically driven p21 overexpression in the liver but ameliorated p21 expression in both the kidney and lung (Extended Data Fig. 8g–i). Using senolytics (either oubaine or navitoclax) resulted routinely in progressive liver damage and adverse mouse outcome. We tested the requirement for renal *Tgfbr1* expression for the renal PTC p21 phenotype using the CCl₄ model. Genetic *Tgfbr1* deletion in the renal[20] and/or hepatic epithelia showed that renal TGFβR1 epithelial expression is required for the renal PTC p21 phenotype resulting from liver disease (Fig. 5j–l). We, therefore, conclude that inhibition of the TGFβ signalling pathway prevents the systemic transmission of senescence to and dysfunction in the kidney.

## Discussion

The SASP is a central mediator of the non-autonomous effects of senescent cells. Here, we demonstrate that senescence can be transmitted to and affect the function of distant organs in a systemic manner. In the context of acute injury, senescence has often been described as part of a finely tuned mechanism with overall beneficial effects for wound healing[21,22]. SASP factors have been shown to induce reprogramming in neighbouring cells, facilitating tissue regeneration[23–25]. However, following severe injury, this mechanism may have the opposite effect, systemically, through excessive SASP production, including senescence itself. In turn, this excessive stimulus for senescence can be associated with compromised organ function.

Systemic transmission of senescence may be relevant to several diseases. Here, we use a series of models of hepatocyte-specific senescence to model an acute senescence phenotype, such as the one observed during ALF. ALF is, itself, characterized by sequential multi-organ failure, typically beginning with the kidney progressing to also involve the brain, lungs and other organs. This clinical progression may, at least in part, be underpinned by the systemic transmission of senescence. The data in patients with acute indeterminate hepatitis, showing that increased hepatocellular expression of p21 at initial presentation before multi-organ failure can predict ensuing multi-organ failure, requirement for liver transplant and/or death, provide evidence for a biomarker that both allows early risk stratification and selection of patients for specific therapies. Similarly, the observation that TGFβ signalling is a central driver of systemic transmission of senescence may pave the way for therapeutic approaches in diseases where this phenomenon occurs. Whilst this effect may either be independent of solely p21-dependent senescence or a phenomenon related to TGFβ activity outside of senescence, it is in line with the beneficial effects of senolytics and senomorphics that have been elegantly demonstrated on numerous pathologies[5,26–28]. Further research is required to dissect out the direct causal link between senescence in the primary tissue and the systemic effects and how they are affected by factors such as disease site or specific senescence phenotype, chronicity of senescence and the interaction with concurrent or pre-existing senescence in other organs. Our results demonstrate that systemic transmission of senescence can induce systemic organ dysfunction, which may be central to multi-systemic sequelae in many diseases.

## Online content

Any methods, additional references, Nature Portfolio reporting summaries, source data, extended data, supplementary information,

acknowledgements, peer review information; details of author contributions and competing interests; and statements of data and code availability are available at https://doi.org/10.1038/s41556-024-01543-3.

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

¹Cancer Research UK Scotland Institute, Garscube Estate, Glasgow, UK. ²School of Cancer Sciences, University of Glasgow, Glasgow, UK. ³Newcastle Fibrosis Research Group, Biosciences Institute, Faculty of Medical Sciences, Newcastle University, Newcastle upon Tyne, UK. ⁴MRC Centre for Inflammation Research, The Queen's Medical Research Institute, University of Edinburgh, Edinburgh, UK. ⁵Kidney Health Service, Royal Brisbane and Women's Hospital, Brisbane, Queensland, Australia. ⁶Department of Hepatology and Gastroentewrology, Campus Virchow-Klinikum, Charité – Universitätsmedizin Berlin, Berlin, Germany. ⁷Liver Failure Group, Institute for Liver and Digestive Health, Division of Medicine, University College London, London, UK. ⁸Fibrofind ltd, William Leech Building, Medical School, Newcastle University, Newcastle upon Tyne, UK. ⁹Medical Toxicology Centre, Edwardson Building, Newcastle University, Health Innovation Neighbourhood, Newcastle upon Tyne, UK. ¹⁰Bioscience, Early Oncology, AstraZeneca, Cambridge, UK. ¹¹Department of Cellular Pathology, Royal Free London NHS Foundation Trust, London, UK. ¹²UCL Cancer Institute, London, UK. ¹³Biosciences Institute, Faculty of Medical Sciences, Newcastle University, Newcastle upon Tyne, UK. ¹⁴Department of Hematology, Mayo Clinic, Rochester, MN, USA. ¹⁵Robert and Arlene Kogod Center on Aging, Mayo Clinic, Rochester, MN, USA. ¹⁶Department of Biochemistry and Molecular Biology, Mayo Clinic, Rochester, MN, USA. ¹⁷European Foundation for the Study of Chronic Liver Failure, Barcelona, Spain. ✉e-mail: t.bird@crukscotlandinstitute.ac.uk

## Methods

### Ethics

This study was conducted in accordance with all relevant ethical regulations. The clinical study was approved by the London-Hampstead Research Ethics Committee (07/Q0501/50) and was in accordance with the declaration of Helsinki as reported before[29]. This study reported multi-lobular necrosis as a predominant histopathological feature being significantly more frequent in non-survivors[29]. All mouse experiments were carried out in accordance with the UK Home Office regulations (licence 70/8891; protocol 2 or licence PP0604995; protocols 3 and 4 or licence P3F79C606, protocol 2) following ethical approval by the local Animal Welfare and Ethical Review Body at the University of Glasgow.

### Patient selection and data collection

The study included 34 consecutive patients with severe acute indeterminate hepatitis who were admitted to a single hospital and underwent transjugular liver biopsy or liver transplantation. The participants were not compensated for participation. Retrospective anonymized data and samples were obtained, following ethical approval and consistent with the UK Human Tissue Act, without informed consent. Of the 34 patients, 14 underwent liver transplantation, and 3 died within 3 months. They were defined as non-survivors. Seventeen patients who recovered spontaneously were defined as survivors. All 34 liver tissue biopsies analysed in this study were obtained at the time of enrolment (baseline biopsies). The following clinical data were collected from the patients: sex, age, date of histopathological examination, history of chronic disease and results of biochemical tests at baseline and at follow-up up to 28 days or the last value before death. These include serum levels of ALT, aspartate transaminase, bilirubin, ALP, international normalized ratio, creatinine, prothrombin time, albumin and hepatic encephalopathy.

### Animal studies

Mice were bred and housed in a licensed, pathogen-free facility, under a 12 h light–dark cycle, at stable temperature (19–23 °C) and humidity (55 ± 10%). The mice were bred on a mixed C57Bl6 background, were housed in conventional cages and had ad libitum access to food and water (standard CRM(E) chow; Special Diets Services no. 801730). All experiments were performed on 8–12-week-old male and female mice, according to the guidelines of the Animal Welfare and Ethical Review Body, and are reported in agreement with the ARRIVE guidelines[30].

Treatment with AAV was performed as described previously[12]. Briefly, mice were injected with either AAV8.TBG.PI.Cre.rBG (AAV-Cre) (Addgene, 107787-AAV8) or AAV8.TBG.PI.Null.bGH (AAV-Null) (Addgene, 105536-AAV8) at a dose of $2 \times 10^{11}$ genomic copies (GC) ml$^{-1}$ unless otherwise stated, in a final volume of 100 µl sterile PBS via a single tail vein injection. Mice in this study weighed 21.9–31.9 g at the time of induction.

For the mouse model of hepatocyte-specific inactivation of Mdm2, male mice homozygous for the Mdm2tm2.1Glo allele (ID: MGI2385439176 (ref. [31])) and the Gt(ROSA)26Sortm14(CAG-tdTomato)Hze allele (ID: MGI3809524177 (ref. [32])) were used (Mdm2$^{E5/E6fl}$; R26$^{LSL-tdTomato/LSL-tdTomato}$ mice). This model was also crossed to create a p21$^{KO}$ model (allele ID: MGI1888950 (ref. [33])). For the induction of a RAS-induced liver senescence, 8–12-week-old male and female mice heterozygous for the Kras$^{tm4Ty}$ allele (ID: MGI:2429948178 (ref. [34])) and the Gt(ROSA)26Sortm14(CAG-tdTomato)Hze allele (ID: MGI3809524177 (ref. [32])) were used (Kras$^{LSL-D12D/WT}$; R26$^{LSL-tdTomato/LSL-tdTomato}$ mice). Epithelial TGFBR1 deletion experiments used 1 µl g$^{-1}$ of CCl$_4$ diluted 1:4 CCl$_4$ in corn oil administered day 3 in mice with alleles AhCre and TGFBR1$^{fl}$ (IDs: MGI3052655 (ref. [35]) and MGI2388050 (ref. [36]), respectively). Mice with AhCre$^{+/-}$ TGFBR1$^{fl/fl}$ were used for epithelial induction utilizing β-naphthoflavone (β-NF; Sigma) versus β-NF-treated AhCre$^{-/-}$ Tgfbr1$^{fl/fl}$ controls, while AhCre$^{-/-}$ TGFBR1$^{fl/fl}$ mice were treated with either AAV-TBG-Cre or AAV-Cre-Null as described above at $2 \times 10^{11}$ GC per

mouse. Genetic controls used both AhCre$^{-/-}$ Tgbfr1$^{fl/fl}$ controls given respective controls (β-NF vehicle or AAV-Cre-Null) and AhCre$^{-/-}$ TGF-BR1$^{fl/fl}$ mice given no induction agent but time matched CCl$_4$ at day 3. The acute CCl$_4$ model used 1 µl g$^{-1}$ of 1:4 CCl$_4$ in corn oil administration as previously described[7]. For treatment with TGFβR1i, the mice received either TGFβR1i or vehicle (0.5% hydroxypropyl methylcellulose (HPMC) and 0.1% Tween 80) by oral gavage twice daily[19]. The dose of TGFβR1i was 50 mg kg$^{-1}$ in 100 ml PBS[7]. Senolytic and senomorphic agents were administered separately; navitoclax (ABT263), oubaine and rapamycin (or vehicle controls) were administered at 100 mg kg$^{-1}$ (gavage twice weekly), 1 mg kg$^{-1}$ (intraperitoneal injection (i.p.) on days 1 and 3 or twice weekly) and 250 µg (i.p. daily), respectively, were given from days 1 to 7 in the ΔMdm2$^{Hep}$ model and day 7 in the Recovery-Mdm2 model. Rapamycin mice in the ΔMdm2$^{Hep}$ model were collected at day 4. All mice receiving either ABT263 or oubaine either deteriorated clinically and were humanely killed or died shortly (within 6 h) after receiving the senolytics. β-NF was given by 3 × 80 mg i.p. injections for epithelial induction in mice with the AhCre allele as previously described[37].

The mice were killed by CO$_2$ inhalation in a CO$_2$ chamber, cervically dislocated and then weighed. Blood was collected immediately via cardiac puncture for whole blood analysis (EDTA buffer-coated tubes; Sarstedt) and plasma biochemistry (lithium–heparin coated tubes; Sarstedt). The plasma was separated by centrifugation (2,350g for 10 min at room temperature) within 1–3 h after culling and stored immediately at −80 °C. After weighing the liver, the caudate and left median lobes lobe were snap frozen on dry ice for protein and RNA extraction and for histology studies, respectively. The remaining liver was fixed in 10% neutral buffered formalin (NBF) for 22–24 h before transfer to 70% ethanol for further processing. The left kidney was immediately cut in half, and both halves were snap frozen on dry ice for protein and RNA extraction and for histology. The right kidney, heart, brain and lungs were fixed in 10% NBF for 22–24 h and then changed to 70% ethanol.

### Assessment of cognitive function

**Y-maze test.** The mice were individually placed into a testing area measuring 25.66 cm × 17.53 cm onto a Samsung Galaxy Tab 2 10.1 for 5 min. Steps and distance travelled were recorded using MouseTrapp software. The Y-maze arena had three arms of 40 cm identified as A1 A2 A3, each with a differentiating marker at the end of the arm. The mice were assigned different start arms in a rotating allocation and were tested before the start of the experiment and on day 4, with differing arm allocations each time. In the T1 phase, the mice were placed into the maze with only two arms open for 5 min to explore the arena. The mice were then removed to a clean cage for 1 min (fresh cage used per cage of mice). Then, in the T2 phase, the mice were returned to the maze in the starting position with the novel arm opened for 2 min and allowed to explore. A camera was set up above the maze to record movements and the video files analysed via Ethovision XT13 software.

**Brain slice electrophysiology.** The animals were humanely killed by anaesthetic overdose with inhaled isoflurane and intramuscular injection of ketamine (≥100 mg kg$^{-1}$) and xylazine (≥10 mg kg$^{-1}$) as previously described[38]. The mice were then transcardially perfused with at least 25 ml of sucrose-rich artificial cerebrospinal fluid—composed of 252 mM sucrose, 3.0 mM KCl, 1.25 mM NaH$_2$PO$_4$, 24 mM NaHCO$_3$, 2.0 mM MgSO$_4$, 2.0 mM CaCl$_2$ and 10 mM glucose. The brain was removed and sliced at 450 µm horizontal slices with a Leica VT1000S vibratome in ice-cold sucrose-rich artificial cerebrospinal fluid. The slices were trimmed to the hippocampal region and maintained at 32–34 °C at an air–liquid interface between normal artificial cerebrospinal fluid (sucrose replaced with 126 mM NaCl) and humidified 95% O$_2$/5% CO$_2$. Oscillations were evoked with 10 µM cholinergic agonist carbachol, to activate transmission through acetylcholine receptors. Extracellular recording electrodes were filled with normal artificial cerebrospinal fluid (resistance 2–5 MΩ), and field recordings taken

from the border between stratum radiatum and stratum lacunosum moleculare in CA3. The recordings were taken with an Axoclamp-2B amplifier (Axon Instruments) and extracellular data filtered at 0.001– 0.4 kHz using Bessel filters. Mains noise was deducted with a Humbug (Digitimer) and data redigitized at 10 kHz using an ITC-16 interface (Digitimer). Axograph 4.6 software (Axon Instruments) was used for data acquisition and analysis.

To generate power spectra Axographs we used Fourier analysis using 60 s per 10 min recording. This was used to calculate peak frequency and area power (area under the peak). The mouse gamma frequency oscillation was measured at frequencies between 15 and 49 Hz. The oscillations were categorized as stable when area power measured within 10% for three consecutive 10 min recording intervals.

### Histology

**Murine samples.** Formalin-fixed, paraffin-embedded sections 4 μm thick were used for simple IHC and for multiplex immunofluorescence. The sections were subjected to heat-induced antigen retrieval, followed by protein blocking to reduce non-specific staining. Incubation with primary antibody overnight at 4 °C or for 1 h at room temperature was followed by secondary antibody incubation and signal detection. The details on the antibodies can be found in Supplementary Table 3.

Photos were taken with a Zeiss Axiovert 200 microscope using a Zeiss Axiocam MRc camera. The stained slides were scanned using a Leica Aperio AT2 slide scanner (Leica Microsystems) at 20× magnification. Automated quantification of positively stained cells or area was performed using the HALO image analysis software (V3.1.1076.363, Indica Labs). Manual quantification of p21+ and BrdU+ kidney cells was performed on 20 random fields at 20× magnification. For p21 IHC quantification on brain sections, total brain area was calculated using the HALO software, and the p21+ cells were manually counted in the whole brain tissue area.

For multiplex immunofluorescence, 4 μm tissue sections underwent heat-induced antigen retrieval by boiling (in a waterbath) in citrate buffer (10 mM Na Citrate (Sigma, W302600), 0.05% Tween 20 (Sigma, P1379), pH 6) for 25 min and were subsequently cooled down for 20 min in the retrieval solution. Peroxidase quenching with 3% $H_2O_2$ (Sigma, 95321) was followed by biotin (Invitrogen, 4303) and protein blocking (Abcam, ab64226). The sections were incubated with the primary antibodies either for 1 h at room temperature or overnight at 4 °C, followed by 45 min with the secondary antibodies (conjugated to fluorophor) together with 4,6-diamidino-2-phenylindole (DAPI, 1 mg ml$^{-1}$). Sudan black B was then used to quench autofluorescence. An aqueous mounting solution (DAKO, S1964) was used for mounting.

The anti-p21 antibody required additional signal amplification, which was achieved by using the tyramide signal amplification system. Briefly, after incubation with the anti-p21 primary antibody, the sections were incubated with a secondary anti-rat biotinylated antibody for 30 min, followed by a 30 min incubation with an avidin–HRP (horse radish peroxidase) complex (Vectastain ABC, Vector, PK-7100). Then, the sections were incubated with tyramide signal amplification fluorescein isothiocyanate (FITC; PerkinElmer, NEL741B001KT) for 6 min (in the dark). After that, the sections that were subjected to a 5 min heat-induced antigen retrieval to remove the anti-p21 antibody complex underwent protein blocking and then were incubated with the other primary antibodies, as described above. The images were taken using a Zeiss 710 upright confocal Z6008 microscope. The Opera Phenix scanner (PerkinElmer) was used to scan the stained sections at 20× magnification. For the analysis of scanned sections, the Columbus software (PerkinElmer) was used to identify hepatocytes and to quantify immunofluorescence staining intensity by hepatocyte in 20 random fields of view.

In situ hybridization was performed in an autostainer (Leica Bond R$_x$) using the 2.5 LSx RNAScope kit (Bio-Techne, 322700) according to the manufacturer's instructions. The probes against *Smad7*

messenger RNA (Bio-Techne, 429418), TGFβ1 (Bio-Techne, 407758), TGFβ2 (Bio-Techne, 406188) and TGFβR1 (Bio-Techne, 431048) were used for the detection of the respective mRNA, and PPIB (Bio-Techne, 313918) was used as a positive control of gene expression.

**Patient samples.** Multiplex immunofluorescence staining was performed as previously described[29,39,40]. Formalin-fixed, paraffin-embedded liver samples were deparaffinized and rehydrated in xylene (Roth) and ethanol (Roth). Antigen retrieval was performed with Tris–EDTA buffer (pH 9) or universal antigen retrieval (Abcam) in a water bath at 98 °C for 30 min, followed by a cooling period of 20–30 minutes. The tissues were blocked with 2% normal goat serum (Thermo Fisher Scientific) to prevent non-specific antibody binding. The slides were incubated overnight at 4 °C with primary antibodies diluted in antibody dilution solution (Life Technology) and stained for 30 min with fluorescently labelled secondary antibodies (Supplementary Table 3) together with DAPI nuclear counterstain (Sigma Aldrich). After scanning the entire slide with a Zeiss Axio Observer7, the images were merged, and the background was subtracted. After each run, the antibodies were stripped by using the 2-mercaptoethanol/ SDS method[39], and the staining was repeated in multiple cycles over an 3 day period. Subsequently, all scans were aligned, hyperstacked and concatenated using the plugin FIJI HyperStackReg V5.6. For binary images, cell segmentation was performed using Ilastik software (v1.3.3). Cell identification and counting, as well as fluorescence intensity measurement, were performed using CellProfiler v3.1.9 and plugin FIJI.

### SA β-Gal staining

Staining for SA β-Gal was performed as described previously[41]. Briefly, 10-mm-thick cryosections or cultured cells were fixed in 2% paraformaldehyde or 0.25% glutaraldehyde in PBS for 15 or 5 min, respectively. This was followed by three washes with PBS 1 mM MgCl$_2$ (pH 5.5 or 6 for mouse or human cells and tissues, respectively) and incubation with staining solution (1 mM MgCl$_2$, 0.5 mM $K_3Fe(CN)_6$ 0.5 mM $C_6FeK_4N_6 \cdot 3H_2O$ and 1 mg ml$^{-1}$X-Gal in PBS, pH 5.5 or 6 for mouse or human cells and tissues, respectively) overnight (liver sections and cells) or for 2.5 h (kidney sections). After three washes with PBS, the cryosections were counterstained with eosin and mounted, while cells (on coverslips) were mounted immediately. For quantification, SA β-Gal+ and SA β-Gal− cultured cells were counted in 20 random fields of view.

### RNA extraction

Whole-tissue RNA was extracted using the Qiagen RNeasy kit (Qiagen, 74104), including the optional DNase step, as described previously[12]. Briefly, 20–30 mg of snap-frozen tissue were homogenized in 600 ml buffer RLT and 1% β-mercaptoethanol using the Precellys Evolution homogenizer (Bertin Technologies), and the RNA was eluted in 30 μl RNase-free water. The RNA concentration was estimated with the Nanodrop 2000m and only samples with a 260/280 ratio ≥2 were used for further analysis.

### Complementary DNA generation and qPCR

cDNA was generated using the QuantiTect Reverse Transcription Kit (Qiagen, 205311) according to the manufacturer's instructions from 1 μg RNA. A PTC-200 Gradient cycler (MJ Research) was used to perform the genomic DNA wipeout and reverse transcription steps. A sample-free reaction and a reaction without the reverse transcriptase served as the negative controls. A real-time quantitative polymerase chain reaction (qPCR) was performed with the SYBR Green system (Qiagen, 204145) using a QuantStudio 5 real-time polymerase chain reaction system in a 384-well-plate setting (final reaction volume 10 μl per well). All primers used were purchased from Qiagen, as shown in Supplementary Table 4. Each biological replicate (mouse) was run in triplicate, and the 18S ribosomal RNA (Rn18S) was used as a house

keeping gene for normalization. Relative expression was calculated using the ΔΔCt method.

## Whole-tissue (bulk) RNA-seq

For bulk RNA sequencing (RNA-seq), the RNA was extracted as described above. Briefly, the RNA was tested on an Agilent 2200 TapeStation (D1000 ScreenTape) using RNA ScreenTape and only samples with a RIN >7 were further processed for library preparation. A total of 20 ng ml$^{-1}$of RNA were used to prepare libraries using the TruSeq Stranded mRNA Kit. The Agilent 2200 Tapestation was used to assess library quality and Qubit (Thermo Fisher Scientific) was used to check concentration. The libraries were then run on the Illumina NextSeq 500 using the High Output 75 cycles kit (paired end, 2 × 36 bp cycle, dual index (I5 and I7 Illumina)).

For the bioinformatics analysis, raw data quality checks and trimming were performed using FastQC (versions 0.11.7 and 0.11.9 for Mdm2 and Kras, respectively), FastP (v0.19.3/0.20.1) and FastQ Screen (v0.12.0/0.14.0). The reads were aligned to the mouse genome and annotation (GRCm38.92 version) using HiSat2 version 2.1.0183. Determination and statistical analysis of expression levels was done by a combination of HTSeq version 0.9.1184, the R environment (v3.4.4/4.1.2), utilizing packages from Bioconductor (v3.6/3.15) and differential gene expression analysis based on the negative binomial distribution using the DESeq2 package (v1.18.1186). Pathway analysis was performed using MetaCore (v2018/2022) from Clarivate Analytics (https://portal.genego.com/).

## scRNA-seq on kidneys

Three ΔMdm2$^{Hep}$ and three control mice were culled by $CO_2$ inhalation; the blood samples were collected by cardiac puncture, and 30–40 ml cold PBS were used to perfuse the circulatory system via the left heart. The six mice were culled and their kidneys processed on two different days. Two mice (one ΔMdm2$^{Hep}$ and one control) were culled together on one day, and the others (two ΔMdm2$^{Hep}$ and two control) were culled 2 months later. On each occasion, the left kidney was used for dissociation and generation of single-cell suspension, while the right kidney was fixed in 10% NBF. The renal capsule of the left kidney was removed, and the kidney was cut into equally sized pieces and was dissociated using a multi-tissue dissociation kit 1 (Miltenyi, 130-110-201) as per the manufacturer's instructions. A total of 0.25 g of kidney tissue were placed in a GentleMACS C tube with dissociation buffer (2.35 ml serum-free RPMI (Roswell Park Memorial Institute) culture medium (Gibco), 100 μl enzyme D, 50 μl enzyme R and 12.5 μl enzyme A). Kidney dissociation was performed in a GentleMACS dissociator using the 'heart_01_01' programme (15 s). The samples were placed in a waterbath (37 °C) for 30 minutes and then placed back in the dissociator ('heart_01_02' programme, 30 s).

After the second round of dissociation, 8 ml of sterile PBS and 10% FBS were added in the C tubes to stop the reaction. The samples were passed through a 40 μm cell strainer into a 50 ml falcon tube. All subsequent steps were performed on wet ice or at 4 °C. The samples were spun at 300 g for 5 min and resuspended in 5 ml of cold PBS. They were spun again at 300 g for 5 min, resuspended in 1 ml red blood cell lysis buffer (8.29 g $NH_4Cl$, 1 g $KHCO_3$ and 37.2 g $Na_2EDTA$ in PBS) and incubated for 30 s on wet ice. The samples were topped with 9 ml of cold PBS and washed twice, as described previously (spin at 300 g for 5 min and resuspended in PBS). The samples were resuspended in cold PBS and were subjected to debris removal using the debris removal solution (Miltenyi) according to the manufacturer's instructions. Finally, the samples were resuspended in 10 ml cold PBS, 10% FBS and 2.5 mM EDTA.

Cell viability and concentration were determined using the Trypan Blue assay and flow cytometry (viability ≥90%). A total of 20,000–40,000 cells were loaded on a 10x Chromium chip (one sample per lane). Cleanup, reverse transcription, cDNA amplification and library preparation were performed using the Chromium Single Cell 3′ Reagent Kits (v3), as per the manufacturer's instructions.

The samples were sequenced in the Illumina NextSeq 500 using the 2 × 150 bp kit with the following read length parameters: 26 bp Read1, cell barcode and Unique Molecular Identifier (UMI); 8 bp I7 index, sample index; and 98 bp Read2, transcript read. CellRanger v.4.0 (with default parameters) was used to demultiplex Illumina BCL output files and align reads to the ensemble GRCm38.99 reference genome with the addition of the AAV8-TBG-Cre and AAV8-TBG-Null sequences.

All other steps were performed using R v.4.0 and packages from Bioconductor v.3.12. The raw matrices generated by CellRanger (v.4.0) were transformed to SingleCellExperiment objects and they were filtered to exclude empty droplets using the DropletUtils v.1.10 package. SingleCellExperiment objects were merged together to perform further downstream analysis. Following similar parameters used in a previous scRNA-seq analysis murine kidney[42], cells with <75 or >3,000 expressed genes or >50% mitochondrial gene expression were filtered out, and only genes expressed in more than ten cells with at least one UMI were kept for further analysis.

The normalization by deconvolution method designed by Lun et al.[43] was used to normalize and log transform the counts with functions from Scater package v.1.18. Highly variable genes were computed with default parameters, and the top 10% were used to perform a principal component analysis and UMAP using the top 20 PCs. After examining UMAP plots coloured by batch run, it was determined that batch correction was not required.

Initial clustering was performed with functions from the Seurat package v.4.0 with default parameters and with eps value of 0.5 and resolution sequence of 0.1 to 1 by 0.1. Markers of each cluster were identified by performing a pairwise differential analysis between each pair of clusters with a minimum difference of 20% of cells expressing the gene and log$_2$ fold change threshold of >0.25 and only keeping the differentially expressed markers in all the comparisons. Reclustering of control cells with a new principal component analysis and UMAP was performed, and the main markers were used to manually classify clusters into different cell types. ΔMdm2$^{Hep}$ cells were projected onto the reference UMAP and assigned to the identified clusters. The same methodology was followed for the identified control PTCs, DTCs and the initial cluster 'mesenchymal cells' for the projection and assignment of the ΔMdm2$^{Hep}$ cells. Scores for senescence, p21, proliferation, TGFβ, JAK–STAT and regeneration signatures were computed using the AddModuleScore Seurat function. The genes included in each list used to create the signatures can be found in Supplementary Table 5. All transcriptomic data will be made publicly available on the Gene Expression Omnibus (GSE189726) repository at the time of publication. Alternatively, cells expressing senescent features were identified by using an unbiased cell type classifier 'singleR', trained on a reference transcriptomic dataset of previously validated murine renal scRNA-seq data[14]. The hyperparameters used were threshold of 0.4, 150 differentially expressed (DE) genes, quantile of 0.6 and fine-tune parameter set to true.

Differential gene expression (DGE) analysis was performed as described by Giustacchini et al.[44]. Briefly, the log$_2$ fold change was calculated between groups, and a non-parametric Wilcoxon test was used to compare the expression values. Fisher's exact test was used to compare expressing cell frequency (percentage of cells per group with at least one UMI count). The P values from both tests were combined using Fisher's method and adjusted using Benjamini–Hochberg. The genes were considered to be differentially expressed if the P adjusted values were <0.05. The heat maps comparing relative gene expression by cell groups were computed using the 'DoHeatmap' function from Seurat R package or 'plotGroupedHeatmap' function from the scater R package. In both cases, the scaled value from each group was calculated by substracting the average logcount gene expression from the total mean gene expression and divided by the standard deviation. A two-colour range scale was used to convert scale values into colour intensity. Gene Set Enrichment Analysis was performed in an unsupervised manner using the 'gseGO' function from the ClusterProfiler

R package; enrichment scores and *P* values were calculated by the function using an empirical phenotype-based permutation test.

## Protein extraction and western blotting

Snap-frozen tissue was homogenized in 300 µl of protein extraction buffer (50 mM HEPES, 100 mM NaF, 150 mM NaCl, 10 mM $Na_4P_2O_7$, 10 mM EDTA, 1% Triton X100, 0.1% SDS and 0.5% Na deoxycholate in ddH2O), 1:100 protease inhibitor (Thermo Fisher, 1862209) and 1:100 phosphatase inhibitor (Thermo Fisher, 1862495) in ddH$_2$O using the Precellys Evolution homogenizer (Bertin technologies). The lysate was passed through a 25G needle five to eight times and was then spun twice at 20,800*g* for 10 min at 4 °C. The Bradford assay was used to measure protein concentration using the Coomasie Plus reagent (Thermo Fisher, 23236) on a 96-well-plate setting. The absorbance was measured immediately at 596 nm and a standard curve was automatically created using a Spectramax reader.

Western blotting was performed on precast gels using the XCell SureLoc Mini-Cell and XCell I Blot Module (Invitrogen, 10572913), and the protein samples were mixed with loading buffer (4× NuPAGE loading buffer (Invitrogen, 11549166) and 5% β-mercaptoethanol) at a concentration of 20 µg µl$^{-1}$. The samples were heated for 5 min at 95 °C and then were spun for 2 min at 20,800*g*. A 20 µl sample and 5 µl of protein ladder (Biorad, 1610373) were loaded on a 4–12%, Bis–Tris, 1.0 mm, ten-well precast gel (Invitrogen, NP0321PK2), and the gel was run at 120 V for 1 h and 50 min in a MOPS running buffer (Invitrogen, NP-0001). Transfer onto polyvinylidene difluoride membranes was performed by wet transfer, using the NuPAGE transfer buffer (Invitrogen, NP-0006) for 1 h at 30 V. Transfer efficiency was assessed by Ponceau S staining. The membranes were blocked for 1 h with 5% BSA buffer and then incubated with the primary antibody (diluted in 5% BSA) overnight at 4 °C. This was followed by 45 min incubation with the HRP-conjugated secondary antibody and enhanced chemiluminescence incubation for the appropriate amount of time (2 min for pSmad2, pSmad3, Smad2 and Smad3 and 5 s for the β-actin). Visualization of the bands was performed using the Chemidoc imaging system (Biorad) and quantification and densitometry was done with ImageJ.

## ELISA and cytokine arrays

For the detection of cystatin C in murine plasma, the 'mouse/Rat cystatin C' immunoassay kit (R&D Technologies, MSCTC0) was used, according to the manufacturer's instructions. The plate was scanned in a plate reader at 450 nm (wavelength correction was set to 570 nm) within 10 min after assay completion. The standard curve concentration calculations were performed on the 'Myassays' website (https://www.myassays.com/).

Cytokine arrays on murine plasma and tissue samples were performed by Eve Technologies, using the TGFβ1, 2, 3 magnetic bead kit for the measurement of the TGFβ ligands and the Milliplex MAP mouse cytokine magnetic bead panel (discovery assay 'Mouse Cytokine Array/ Chemokine Array 31-Plex (MD31)').

## Metabolomics on mouse urine

For the detection of urine amino acids, urine was collected from the mice by free urination 2 and 3 days before AAV-Cre injection on injection day, as well as 3 and 4 days post AAV-Cre injection. The urine collected by free urination after scruffing the mice was diluted 1:50 in cold metabolite extraction buffer (50% methanol, 30% acetonitrile and 20% water) and were vortexed for 30 s. The samples were then centrifuged at 20,800*g* for 10 min at 4 °C. Liquid chromatography–mass spectrometry was performed as described previously[45]. The source data are presented in Supplementary Table 6.

## Cell culture

**MEFs.** WT MEFs were isolated from one E13-E14 WT C57Bl/6 embryo. The MEFs were cultured on 10 cm Petri dishes (Corning) in Dulbecco's modified Eagle medium supplemented with 10% FBS, 1% penicillin–streptomycin, 1% L-glutamine (cDMEM) at low O$_2$ (3%). The MEFs cultures were confirmed to be free of mycoplasma contamination. For the plasma treatments, passage (P)3–P5 MEFs were trypsinized, and the cell density was determined by using the CASY cell counter (Cambridge Bioscience, 5651808). A total of 30,000 cells (in 1 ml medium) were plated on 24-well-plate wells (with a round coverslip in each well) and were incubated with cDMEM with 1% plasma from either ΔMdm2$^{Hep}$ or control mice for 24 h. The cDMEM with plasma was changed with fresh cDMEM every 2 days, and 6 days after the first medium change, the cells were stained for SA β-Gal as described above. The coverslips were mounted on the slides, and the SA β-Gal$^+$ cells were quantified by manual counting of 20 random fields of view of using a Zeiss Axiovert 200 microscope at 20× magnification.

**NS cell-derived neuronal cells.** Neuronal cells were derived from human foetal neural crest progenitors as previously described[46]. The cells were passaged using trypsin-EDTA solution and were seeded into a geltrex coated 48-well cell culture plate at a density of 5,000 cells per well with 250 µl of proliferation medium. They were allowed to proliferate for 2 days before the proliferation medium was withdrawn and replaced with differentiation medium. The growth medium was replaced with differentiation medium for 16 days before beginning the experimental procedure. The composition of the proliferation and differentiation media has been described previously[47].

For the plasma treatment experiments, plasma samples were thawed on ice, vortexed and heat inactivated for 30 min at 60 °C. They were diluted in media to the desired concentration (1:100) and added to the plate for 24 h. For the additional treatment with TGFβR1i, either TGFβR1i (0.2 mM) or vehicle (dimethylsulfoxide, DMSO) were mixed with the same plasma-containing medium and stayed on the cells for 24 h. Two days later, the cells were stained for SA β-Gal, and the positive cells were quantified as described in 'SA β-Gal staining' section.

## PCLS and PCKS

**Ethics.** Human kidney tissue will be collected from donor kidneys declined for transplantation and accepted for research, through the Newcastle Transplant Tissue Biobank under project IOT054.

**Kidney slice culture.** Eight millimetre human kidney and murine kidney (WT) or liver (*Mdm2fl/fl* or *ΔMdm2$^{Hep}$*) tissue cores were generated using an 8 mm Stiefel biopsy punch, placed in a metal mould and 3% low gelling temperature agarose and allowed to set. The slices were then generated and cultured according to methods previously described[48,49]. Human kidney slices were cultured with 10 ng TGFβ1 to induce fibrosis and treated with ±10 µM TGFβR1i SB-525334 (Sigma Aldrich, S8822, batch number: 000014491).

**Media transfer experiment.** Murine PCLS were generated from liver (*Mdm2$^{fl/fl}$* or *ΔMdm2$^{Hep}$*), and the medium was collected and pooled after 24 and 48 h and concentrated using Amicon Ultra-15 Centrifugal Filter Units (Merck, UFC9003). WT murine kidney slices were cultured in PCKS medium containing 0.5% v/v of concentrated cultured PCLS-conditioned medium (equivalent to 25% PCLS condition medium at final concentration) from the livers of control *Mdm2$^{fl/fl}$* or *ΔMdm2$^{Hep}$* mice, with or without 200 nM AZ12601011.

## Statistical analyses and graphs

The Prism 9 Software (GraphPad Software) was used for statistical analyses. The Shapiro–Wilk test was used to assess data normality. For normally distributed data, the one-way analysis of variance (ANOVA), two-way ANOVA, the Brown–Forsythe and Welch ANOVA test, paired and unpaired *t*-tests or the Welch's *t*-test, were used to test for statistical significance between data groups. The Kruskal–Wallis test or the Mann–Whitney test were performed for non-parametric data.

All statistical tests comparing two groups were two-tailed. All figures were created using the Scribus Software (v1.4.7, G.N.U. general public licence). Unless otherwise stated, all data points on the line or bar graphs represent the mean ± standard error of the mean (s.e.m.), and each dot represents a single mouse.

## Statistics and reproducibility

No statistical methods were used to pre-determine sample sizes, but our sample sizes are similar to those reported in previous publications[3,7,11,12]. For animal experiments, the biological replicate sizes were chosen taking into account the variability observed in pilot and prior studies using AAV-TBG-Cre in the Mdm2 model. For all experiments, the animal/sample assignment was matched for sex and age-matched controls and were assigned based upon randomly assigned mouse identification markings. Batched staining and analyses alongside controls were used throughout. The investigators were not blinded for the in vivo experiments. No animals or data points were excluded from analyses. The technical staff administering the therapy were blinded to the mouse genotypes. All subsequent tissue handling and analysis were blinded and/or performed using standardized automated analyses where possible. Western blot studies were performed without blinding, given that samples were run by condition for visualization. In the figure legends, *n* represents the number of mice, unless explicitly stated. The data distribution for normality and testing of equal variances were assessed using Prism 9 Software. No animals or data points were excluded from analyses.

## Reporting summary

Further information on research design is available in the Nature Portfolio Reporting Summary linked to this article.

## Data availability

Source metabolomics data can be found in Supplementary Table 6. The fastq files and processed data for the scRNA-seq analysis of mouse kidney cells and bulk tissue transcriptomics in liver and kidney can be found on the Gene Expression Omnibus repository (accession numbers GSE189726, GSE267196 and GSE262705). Source data are provided with this paper. All other data generated and/or analysed during the current study are available from the corresponding author on reasonable request.

## Code availability

No custom code or mathematical algorithms were used in this manuscript.

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

## Acknowledgements

We thank the Cancer Research UK (CRUK) Scotland Institute's histological services, biological services and molecular technology, metabolomics and bioinformatics services and central services, as well as C. Winchester (CRUK Scotland Institute) and the Veterinary Clinical Pathology Lab (University of Glasgow), for their help. We thank K. Simpon and J. Davidson at the Scottish Liver Transplant Unit and J. Dyson and E. Burton on behalf of the UK-AIH consortium for access to data in patients with liver disease. C.K., Y.-C.H., W.C., C.N., R.S., D.S., G.M., S.M., E.H.T., T.S. and A.C. were funded by Cancer Research UK (grant number A17196 and A31287). M.T.-T. was supported by Medical Research Council (MRC) funded PhD studentships (MR/N013166/1). L.M.G was funded by a PhD studentship from the National Institute for Health Research Newcastle Biomedical Research Centre. A.G. was funded by Cancer Research UK (A26813). A.L.C. was funded by William Harker Foundation. E.D.O.S. was funded by Queensland Health Targeted Clinical Research Fellowship and Metro North Clinical Research Fellowship, Queensland. D.P.B. was funded by an MRC fellowship (MR/W00089X/1). C.M.M. and P.S.H. were funded an MRC (MR/X004112/1), a Brains for Dementia (BDR-IPSC-001; a joint venture between Alzheimer's Society and Alzheimer's Research UK)

and a National Institute for Health Research (HPRU-2012-10076) grant. The human embryonic material was provided by the joint MRC and Wellcome Trust (MR/R006237/1, MR/X008304/1 and 226202/Z/22/Z) Human Developmental Biology Resource (www.hdbr.org). A.K.N. was supported by the CRUK Rosetta Grand Challenge grant (A25045). M.M. and T.G.B. were funded by the Wellcome Trust (grant number WT107492Z) and CRUK HUNTER Accelerator Award (grant number 175 A26813). G.J.I. was funded by Cancer Research UK (grant number A29802). The programme of work on human samples (M.H., C.E., F.A., A.Q. and R.J.) was supported by a European Union's Horizon H2020 programme grant agreement (945096). K.K. was funded by a John Goldman Fellowship sponsored by Leukaemia U.K. (2019/JGF/003). O.J.S. and T.G.B. were funded by Cancer Research UK (Grant number A21139) and an MRC programme grant (MR/Y003365/1). F.O. and L.A.B. were funded by the MRC grants (MR/K0019494/1 and MR/R023026/1) and F.O. by a Rosetrees Trust project grant (PGL22/100014).

## Author contributions

C.K. contributed to the conceptualization of the project, designed and performed animal studies, designed and performed experiments, analysed data, created the figures and wrote the manuscript (original draft and subsequent editing). M.T.-T., E.D.O. and D.P.B. performed the bioinformatics analysis of the scRNA-seq data and created the figures. L.M.G., A.C., L.H.R., L.A.B. and T.S. performed animal studies and in vitro experiments, including the human and murine slices and analysed data. Y.-C.H. and K.K. performed the scRNA-seq. C.N. and CRUK Scotland Institute's histological services performed IHC staining. W.C. performed the bulk RNA-seq. R.S. performed the analysis of the bulk RNA-seq. D.S., G.M. and T.S. performed the metabolomics experiments. S.M., A.G., E.H.T. and M.M. assisted with animal experiments. A.K.N. assisted with experiments. P.H., C.M.M., F.E.N.L., K.K. and F.O. designed experiments and analysed data. A.C., S.T.B., G.J.I. and O.J.S. provided resources. M.H., C.E., F.A., A.Q. and R.J. collated patient data and samples and performed the human studies in patients with acute indeterminate hepatitis and generated the relevant data. T.G.B. contributed to the conceptualization of the project, assisted with data analysis and figure generation, edited the original draft, provided resources and acquired funding. All authors contributed to the drafting and editing of the manuscript.

## Competing interests

S.T.B. is an employee and shareholder of AstraZeneca. L.A.B. and F.O. are directors, employees and shareholders of Fibrofind, and L.H.R. is an employee and shareholder of Fibrofind. A patent entitled 'Agents for use in a method of treating an acute liver disease' has been filed in relation to the work associating senescence markers with patient outcomes; patent applicants: Beatson Institute for Cancer Research, University College London, Royal Free London NHS Foundation Trust, Charité – Universitätsmedizin Berlin; inventor(s)—T.G.B., F.A., R.J. and A.Q. (application number 2309181.2). R.J. is the inventor of OPA, which has been patented by UCL and licensed to Mallinckrodt Pharma. He is also the founder of Yaqrit Limited, Hepyx Limited (spin out companies from University College London) and Cyberliver. He has research collaborations with Yaqrit Limited. The other authors do not have competing interests to declare.

## Additional information

**Extended data** is available for this paper at https://doi.org/10.1038/s41556-024-01543-3.

**Correspondence and requests for materials** should be addressed to Thomas G. Bird.

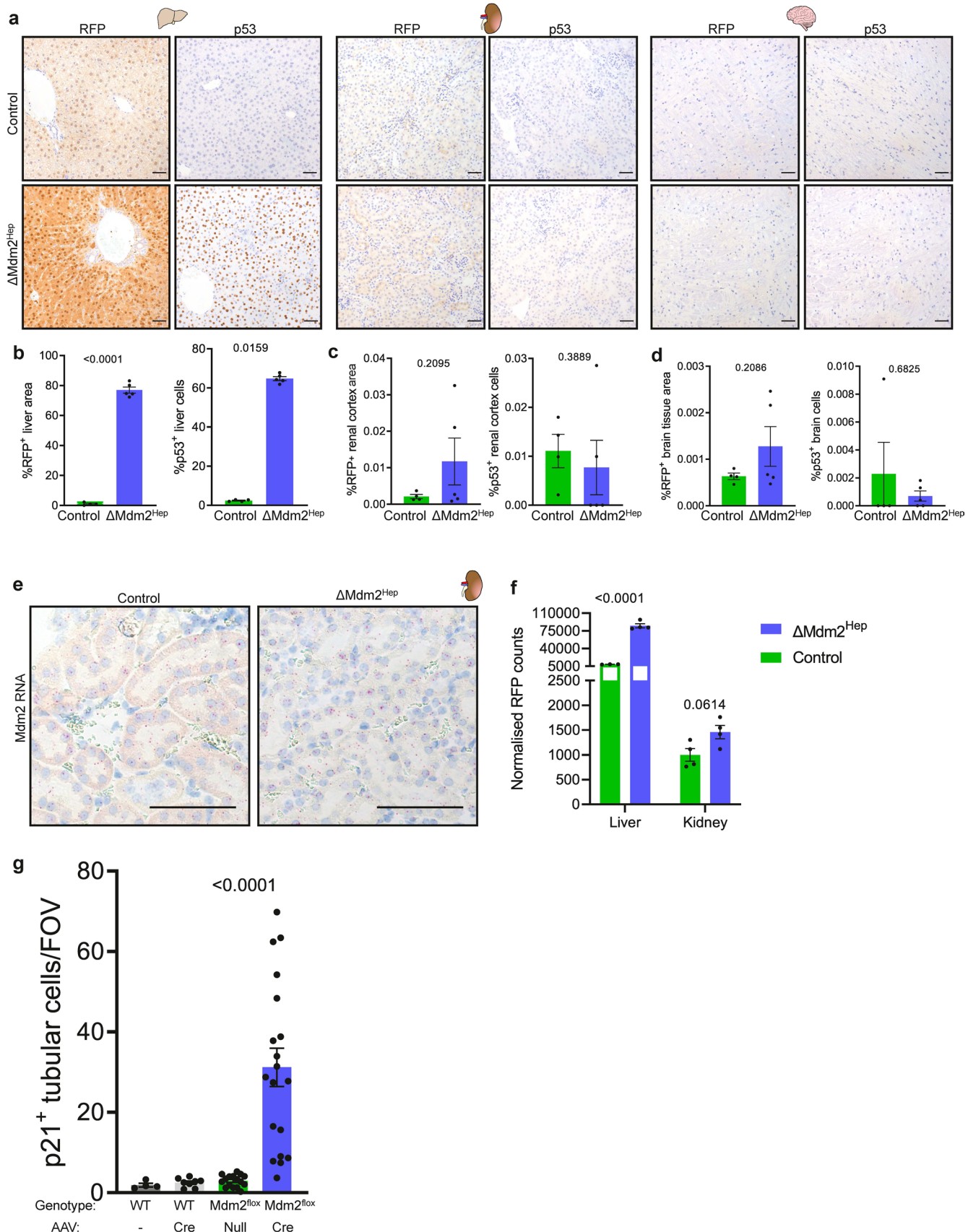

**Extended Data Fig. 1 | See next page for caption.**

**Extended Data Fig. 1 | The AAV8-TBG-Cre vector is hepatocyte-specific and does not result in notable genetic recombination of extra-hepatic tissues.** (**a**) Representative IHC images of liver (left), kidney (middle) and brain (right) sections of control and ΔMdm2^Hep mice stained for RFP and p53; mice were culled 4 days post AAV injection. n=4 and 5 control and ΔMdm2^Hep mice, respectively. (**b**), (**c**), (**d**) Automated quantification of p53⁺ cells and RFP⁺ area on liver, renal cortex and brain tissue respectively, n=4 and 5 control and ΔMdm2^Hep mice, respectively, in all graphs. For panel (**b**), unpaired two-tailed t-test (RFP) and Mann-Whitney test (p53) were used. For panels (**c**) and (**d**), two-tailed Welch's t-test (RFP) and two-tailed Mann-Whitney test (p53) were used. (**e**) Representative images from ISH-stained kidney sections with a probe that specifically detects the junction of murine *Mdm2* exons 5 and 6. (**f**) Normalised RFP transcript counts

(FPKMs) from bulk liver and bulk kidney RNA sequencing data. n=4 except n=3 in liver control; unpaired two-tailed t-test. (**g**) Manual quantification of P21⁺ renal tubular cells from either 8–12 weeks old male WT or male *Mdm2*^E5/E6fl; *R26*^LSL-tdTomato/LSL-tdTomato (Mdm2^flox) mice, untreated (-) or intravenously injected with either 2x10^11 GC/mouse AAV8-TBG-Cre (Cre) or with AAV8-TBG-Null (Null) and culled 4 days later: additional control data from Fig. 1e are presented as p21⁺ tubular cells per field of view (FOV), n=4 and 8 for untreated vs Cre in WT mice and n=16 and 19 Null and Cre treated ΔMdm2^Hep mice; one way Anova, adjusted p value denotes Mdm2 Cre vs Null; all other comparisons p>0.99. Throughout all bars are mean ± S.E.M and the numbers on the graphs are p values. Scale bars are 50μm. Source numerical data are available in source data.

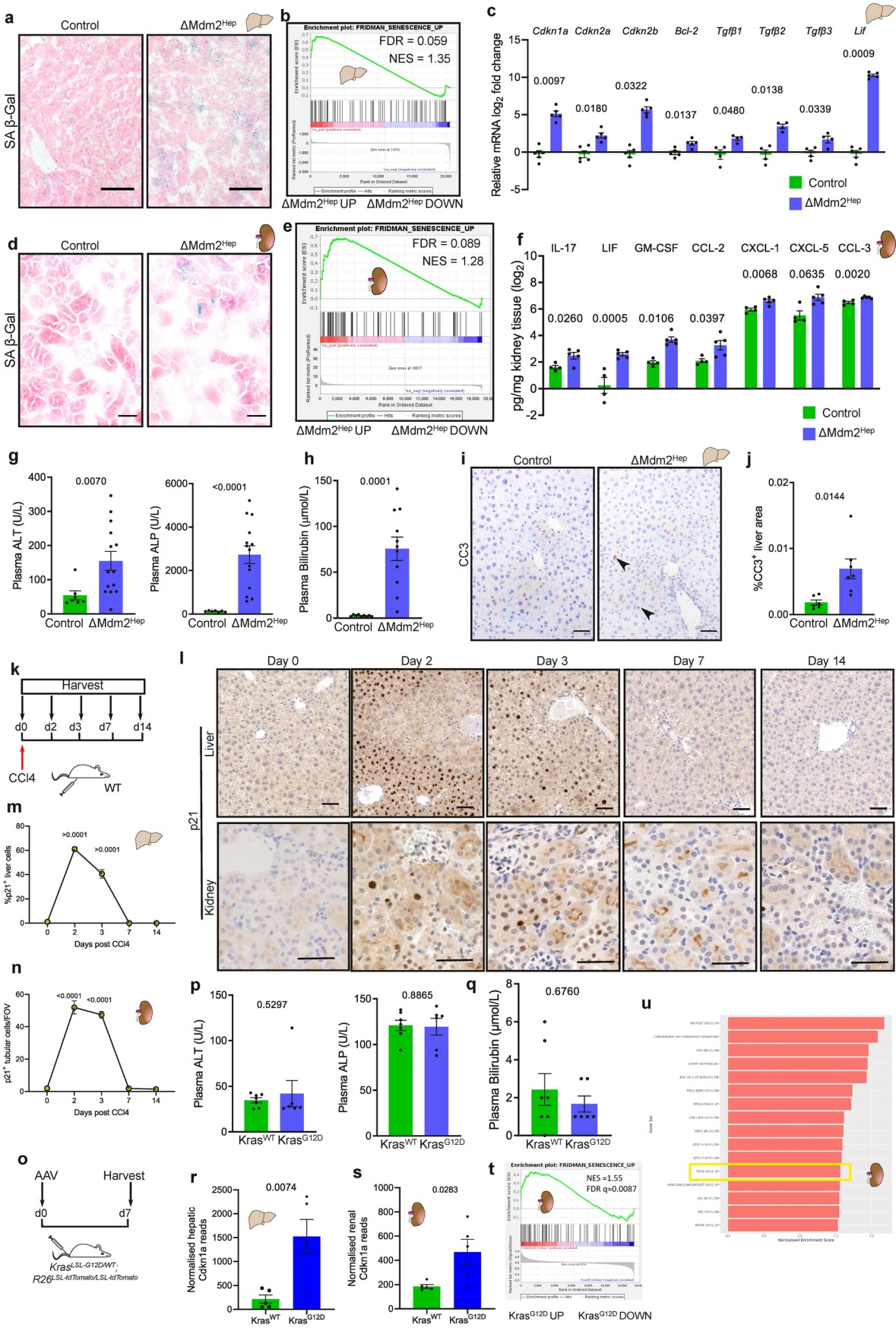

**Extended Data Fig. 2 | See next page for caption.**

**Extended Data Fig. 2 | Liver senescence is associated with renal senescence.**
(**a**) Representative images of control and ΔMdm2[Hep] liver sections stained for SA β-Gal; n=8 mice per group. (**b**) Targeted Gene Set Enrichment Analysis (GSEA) for a senescence signature on the whole liver RNA-seq dataset performed on whole liver RNA isolated from day 4 kidneys in the ΔMdm2[Hep] model and its control. (**c**) RT-qPCR analysis of *Cdkn1a*, *Cdkn2a*, *Cdkn2b*, *Bcl2*, *Tgfb1*, *Tgfb2*, *Tgfb3* and *Lif* expression in whole liver lysates in the ΔMdm2[Hep] model. n=4 mice for *Tgfb1*, *Tgfb2* and *Tgfb3 and* n=5 in all others; n=4 in *Tgfbs* due to misloading of one sample. Unpaired two-tailed t test for *Cdkn1a*, *Cdkn2a, Tgfb1, Tgfb3 and Bcl2*; two-tailed Welch's t test for *Tgfb2, Lif* and *Cdkn2b* (unselected samples from larger cohorts; see Fig. 1 for details). (**d**) Representative images of control and ΔMdm2[Hep] model kidney sections stained for SA β-Gal. n=8 mice per group (unselected samples from larger cohorts). (**e**) Targeted GSEA for a senescence signature on the whole kidney RNA-seq dataset performed on whole liver RNA isolated from day 4 kidneys in the ΔMdm2[Hep] model and its control. (**f**) Cytokine arrays on whole kidney lysates for a range of SASP factors in the ΔMdm2[Hep] model. n=4 control mice and n=5 ΔMdm2[Hep] mice (unselected samples from larger cohorts). Unpaired two-tailed t-test (IL-17, LIF, GM-CSF, CXCL-1 and CCL3) or two-tailed Mann-Whitney test (CCL-2, CXCL-5). (**g**) Alanine aminotransferase (ALT) and alkaline phosphatase (ALP) in the plasma of ΔMdm2[Hep] and control mice, n=7 and 14 control and ΔMdm2[Hep] mice (unselected samples from larger cohorts; samples failing quality control were excluded), respectively, for both graphs; two-tailed Mann-Whitney test (ALT) or two-tailed Welch's t-test (ALP). (**h**) Plasma bilirubin in ΔMdm2[Hep] and control mice, n=9 and 11 control and ΔMdm2[Hep] mice respectively (unselected samples from larger cohorts, samples failing quality control were excluded); two-tailed Mann-Whitney test.

(**i**) Representative images of cleaved caspase 3 (CC3) IHC on liver sections of ΔMdm2[Hep] and control mice. Arrowheads highlight CC3[+] cells. (**j**) Automated quantification of CC3 IHC on liver sections: data are presented as CC3[+] area as a percentage of total liver area, n=6 and 7 control and ΔMdm2[Hep] mice respectively (unselected samples from larger cohorts); two-tailed Welch's t-test. (**k**) Schematic of the CCL$_4$ treatment of WT mice. C57Bl/6 mice were injected with CCL$_4$ and were sacrificed 2, 3, 7 or 14 days post injection. (**l**) Representative images of p21-stained liver and kidney sections of CCL$_4$-injected mice. (**m**) Automated quantification of p21[+] liver cells on liver sections or (**n**) manual quantification of p21[+] renal tubular cells on kidney sections of the CCL$_4$-injected mice. n=5 mice for each timepoint except day 0 where n=4; One-way ANOVA. (**o**) Schematic of AAV-mediated induction of the *Kras*[G12D] mice. 8–12 weeks old mice were injected with either AAV-Null or AAV-Cre and were sacrificed 7 days post induction. (**p**) ALT and ALP levels in the plasma of Kras[WT] and Kras[G12D] mice. n=7 Kras[WT] and n=6 Kras[G12D] mice for both ALT and ALP. Two-tailed Mann-Whitney test (ALT) or unpaired two-tailed t-test (ALP). (**q**) Plasma bilirubin in Kras[WT] and Kras[G12D] mice, n=7 Kras[WT] and n=6 Kras[G12D] mice; two-tailed Mann-Whitney test. Normalised liver (**r**) and kidney (**s**) *Cdkn1a* reads in *Kras*[WT] versus *Kras*[G12D] mice from bulk RNAseq; n=5 mice in both groups (unselected samples from the larger cohort); unpaired two-tailed t-tests. (**t**) Targeted GSEA for a senescence signature on the whole RNA-seq dataset of *Kras*[WT] versus *Kras*[G12D] kidneys. (**u**) Top-16 upregulated pathways in an unsupervised GSEA performed on *Kras*[WT] versus *Kras*[G12D] kidneys. For all graphs, mice were culled 4 days post AAV injection. Throughout all bars are mean ± S.E.M. and the numbers on the graphs are p values. Scale bars are 50μm. Source numerical data are available in source data.

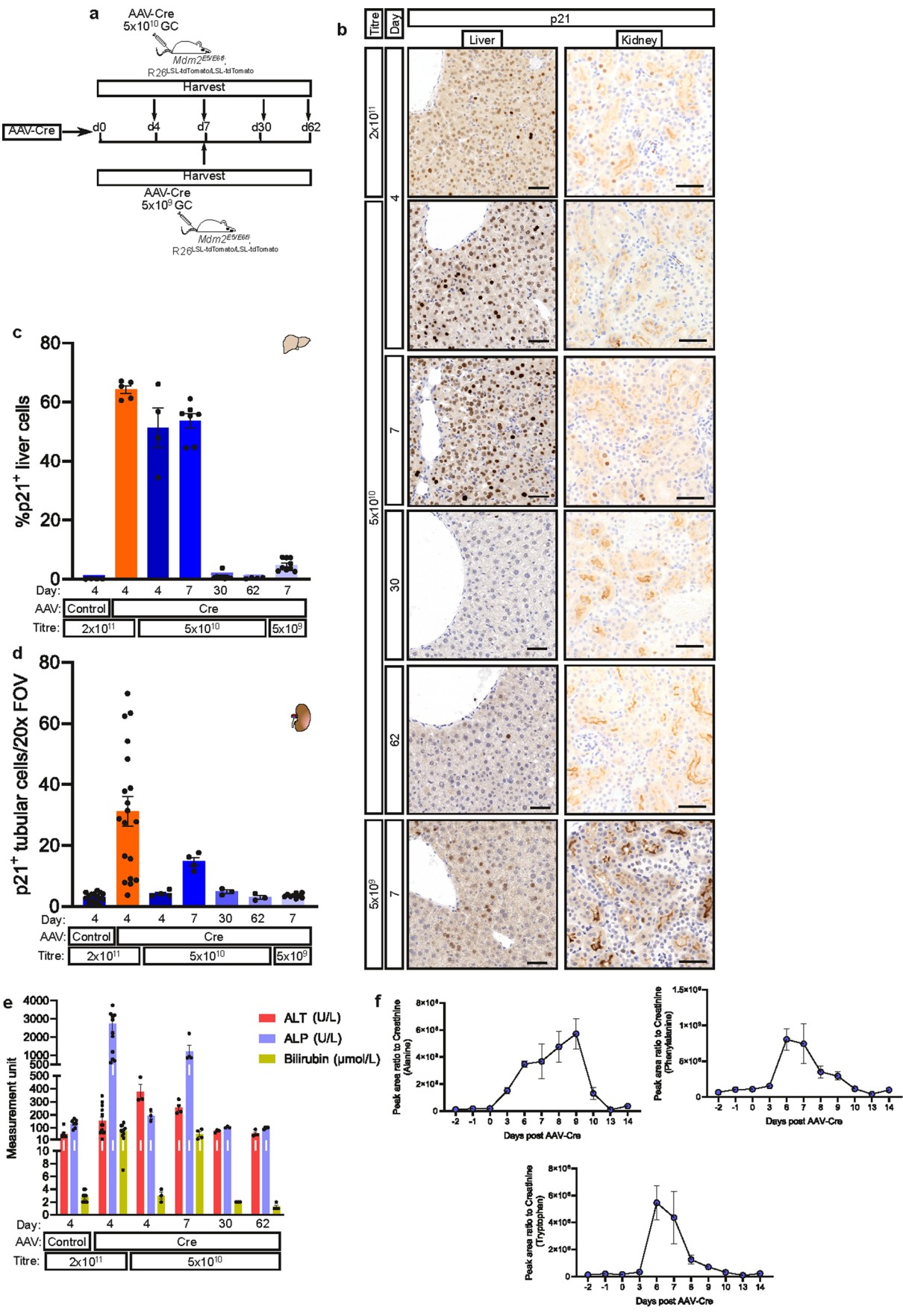

**Extended Data Fig. 3 | See next page for caption.**

**Extended Data Fig. 3 | The intensity of the renal senescence-like phenotype in ΔMdm2$^{Hep}$ mice is proportionate to the level of liver senescence.**
(**a**) Schematic of AAV dose titration in ΔMdm2$^{Hep}$ mice in the recovery-Mdm2$^{Hep}$. 8–12 weeks old male *Mdm2*$^{E5/E6fl}$; *R26*$^{LSL-tdTomato/LSL-tdTomato}$ mice were either injected with 5x10$^{10}$ GC/mouse of AAV-Cre and then harvested 4, 7, 30 or 62 days post injection, or they were injected with 5x10$^{9}$ GC of AAV-Cre and harvested 7 days post injection. (**b**) Representative images of p21-stained liver and kidney sections from the AAV dose titration experiment. (**c**) Automated quantification of p21$^{+}$ liver cells on liver sections from the AAV dose titration experiment; data are presented as percentage of total liver cells. n=4,5,4,7,8,4 and 9 from left to right;; subgroup members of the 2x10$^{11}$ AAV-Cre and controls were analysed, and all member of the recovery model are included. (**d**) Manual quantification of p21$^{+}$

renal tubular cells: data are presented as mean p21$^{+}$ tubular cells per 20 fields of view (FOV); n=16,19, 4,4,3,3 and 9 from left to right; whole cohorts of 2x10$^{11}$ AAV-Cre and controls are included (data for day 4 is shown also in Fig. 1e), the recovery model was performed in smaller cohorts with n= 3-9 with all biological replicates assessed included in analyses. (**e**) Serum levels of ALT, ALP and Bilirubin in the mice of the AAV dose titration experiment. Biological replicates (separate animals) n=7, 14, 3, 4, 3 and 4 for ALT and ALP and 9, 11, 3, 4, 3 and 4 for Bilirubin from conditions from left to right (**f**) Urine levels of Alanine, Phenylalanine and Tryptophan identified by liquid chromatography-mass spectrometry (LC-MS) in ΔMdm2$^{Hep}$ mice pre- and post- AAV-Cre injection: n=4 mice. Throughout all bars are mean ± S.E.M. Scale bars are 50μm. Source numerical data are available in source data.

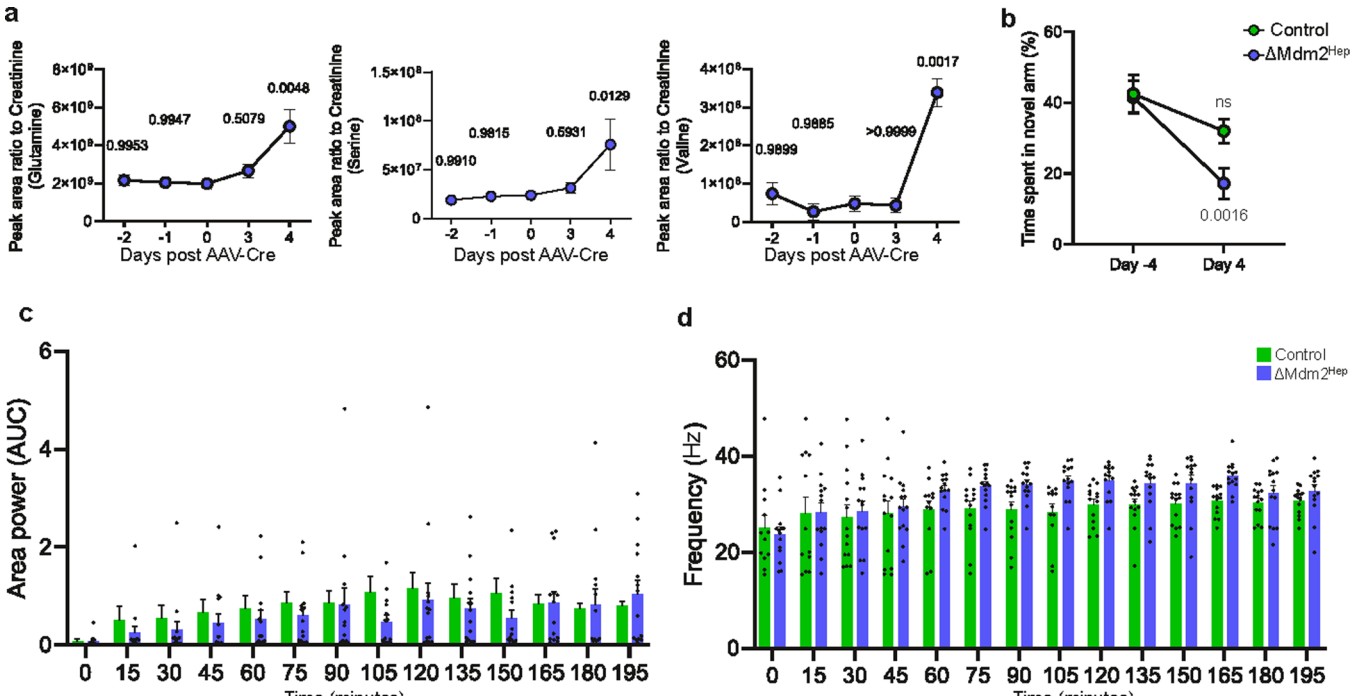

**Extended Data Fig. 4 | Renal and brain dysfunction in response to liver senescence.** (**a**) Urine levels of Glutamine, Serine and Valine identified by liquid chromatography-mass spectrometry (LC-MS) in ΔMdm2^Hep mice pre- and post-AAV-Cre injection: dots represent the average peak area of n=3 mice at days -2 and 0, n=4 mice at day 4 and n=5 mice at days -1 and 3; 2-way ANOVA comparing each time point to induction day (day 0). (**b**) Proportion of time spent by control and ΔMdm2^Hep mice in the novel arm of the Y-maze 4 days before and 4 days after AAV injection: dots represent the average percentage of time spent in the novel arm for n=7 and 9 for control and ΔMdm2^Hep mice respectively; 2-way ANOVA. Area under the curve (area power) (**c**) and frequency (Hz) (**d**) of the brain slice oscillations after stimulation with carbachol. n=14 brain slices (except n=12 at times 0,15,60 and 105) from 4 control mice and n=14 brain slices from 4 ΔMdm2^Hep mice for every time point. Throughout all bars are mean ± S.E.M. and the numbers on the graphs are p values. Source numerical data are available in source data.

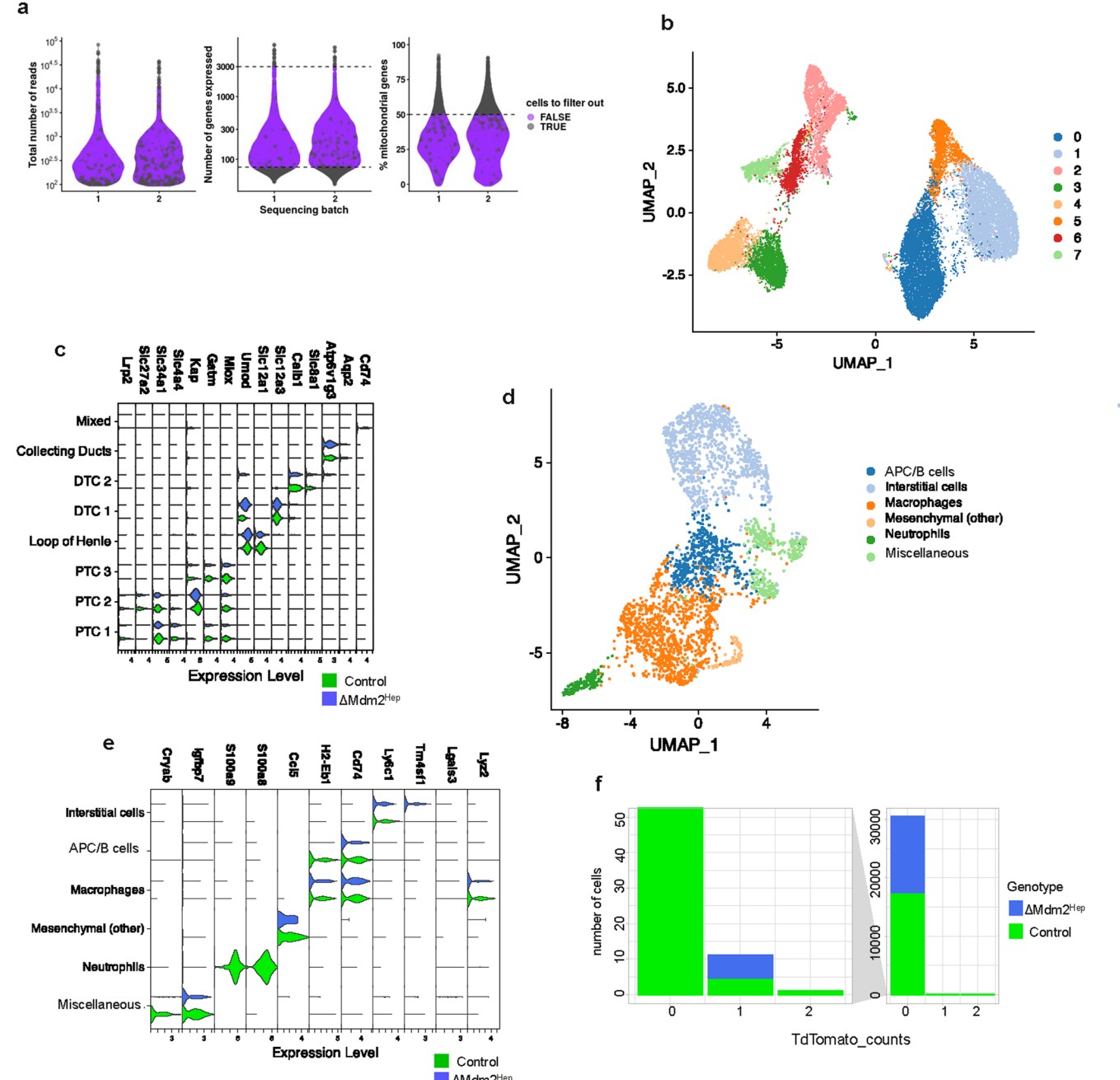

**Extended Data Fig. 5 | Sub-clustering and cell type assignment in the scRNA-seq data. (a)** scRNA-seq analysis of 3 kidneys from control mice and 3 kidneys from ΔMdm2$^{Hep}$ mice taken 4 days after induction. After single cell sequencing cells were filtered for quality based upon the number of genes expressed and percentage of mitochondrial gene reads in each event. Bar charts are shown, by each sequencing batch, total number of reads per cells (left), with cells filtered out shown in grey and those included in analyses shown in purple based upon number of reads (centre; <75 or >3000) and % mitochondrial genes/total (right; >50%). **(b)** UMAP plot of 24,215 single-cell transcriptomes from 6 mouse kidneys. **(c)** Table with violin plots showing the expression levels of marker genes across the 8 clusters. **(d)** UMAP plot of the cells following sub-clustering performed upon the mixed cluster shown in Fig. 3b. **(e)** Table with violin plots showing the expression levels of marker genes across the 6 subclusters within the mixed (#3) cluster. **(f)** Bar plot showing the number of RFP UMIs per cell in control and ΔMdm2$^{Hep}$ cells. Source numerical data are available in source data.

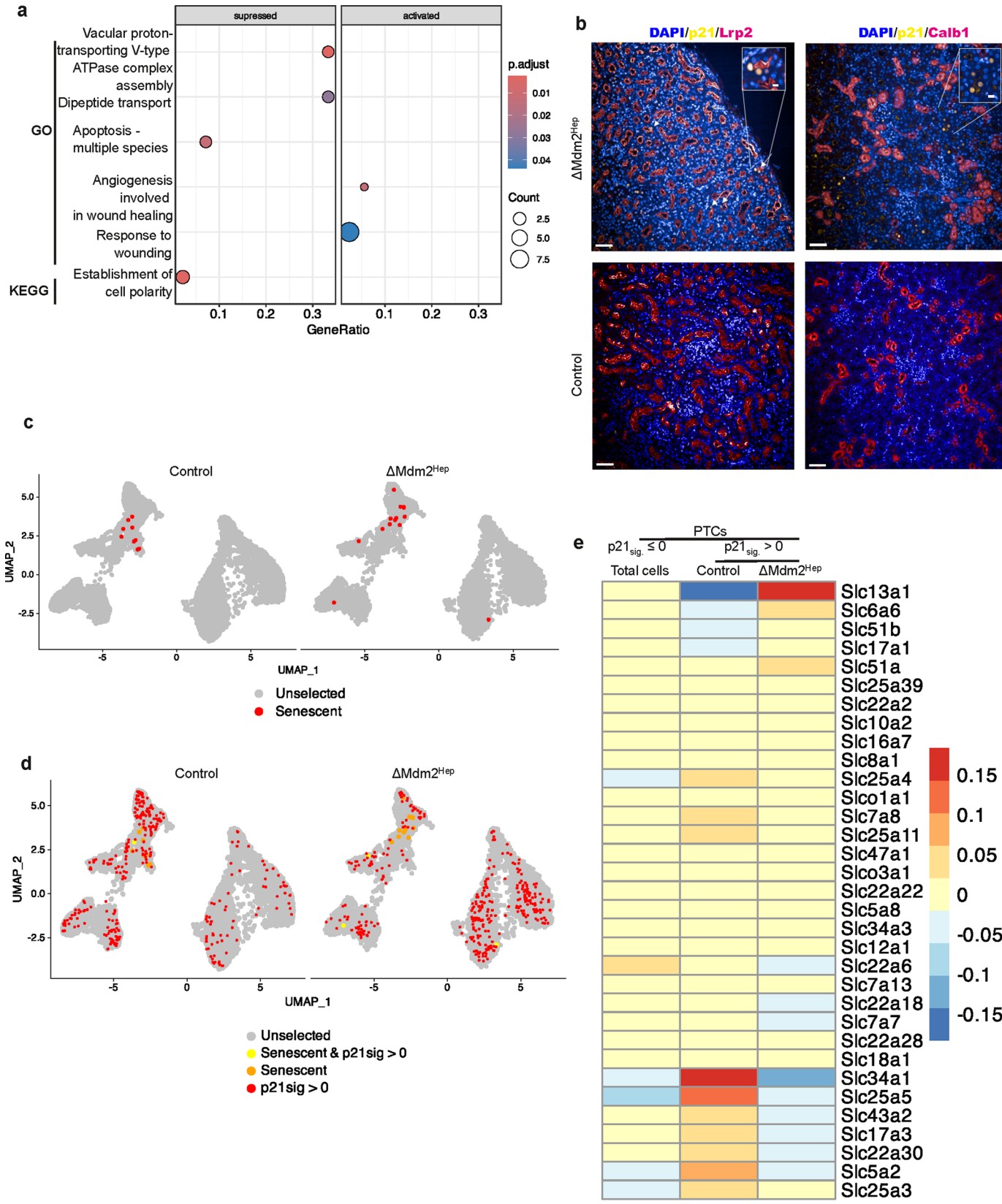

**Extended Data Fig. 6 | See next page for caption.**

**Extended Data Fig. 6 | Liver senescence results in transcriptional changes in the renal proximal tubular cell compartment.** (**a**) Dot plot of pathways resulting from unsupervised GSEA of differentially expressed genes between ΔMdm2Hep and control PTCs. Differential expression analysis was performed between ΔMdm2Hep and control PTCs and the ranked gene set was used to perform unsupervised GSEA against Gene Ontology (GO) and KEGG pathways. The size of the dots represents the number of downregulated (suppressed, left) or upregulated (activated, right) genes in the ΔMdm2Hep PTCs for each pathway (count) with colour representing statistical significance; Permutation test. (**b**) Representative images from a single batch of dual immunofluorescence staining for p21 and renal tubular epithelial markers (LRP2 or CALB1) on ΔMdm2Hep kidneys at day 4; LRP2 is expressed on the apical brush boarder of the PTCs; n-5/6 for control/ ΔMdm2Hep respectively. Scale bars are 50μm and 5μm in inset magnification. (**c**) UMAP plot showing the clusters of the cells (red versus grey) of the scRNA-seq data from analysis of 3 kidneys from control mice and 3 kidneys from ΔMdm2Hep mice taken 4 days after induction highlighting cells with a positive score for the senescence signature (sen) from O'Sullivan et al.[14]. (**d**) UMAP plot of the same cells shown in panel c highlighting co-occurrence (yellow) of the p21 signatures (Fig. 3c) and the renal tubular senescence signature from O'Sullivan et al.[1]. (**e**) Heatmap of relative expression of Slc transporter genes in the PTC compartment grouped by p21 signature score in each experimental condition; PTCs without the p21 signature in both conditions are merged as the left-hand column. Source numerical data are available in source data.

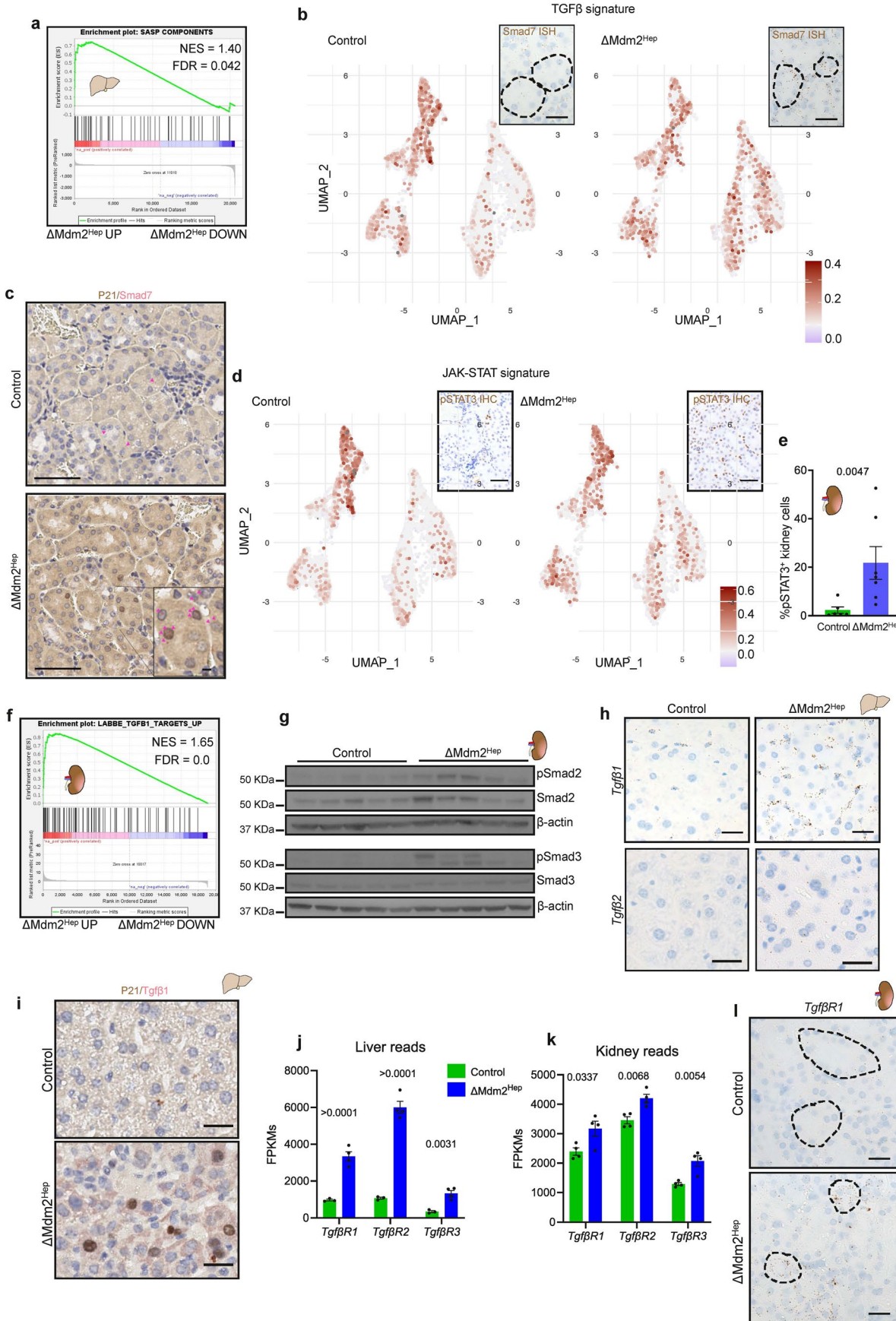

**Extended Data Fig. 7 | See next page for caption.**

**Extended Data Fig. 7 | Liver-derived SASP factors affect the TGFB and LIF/ JAK-STAT signalling pathways in the kidney. (a)** GSEA plot for a SASP gene set on the significant differentially expressed genes from the bulk liver RNA-seq data in the ΔMdm2^Hep model versus control at day 4. **(b)** UMAP plots showing the distribution of the cells that have a positive score for the TGFβ gene signature in the control (left) and the ΔMdm2^Hep model (right) renal cells. Inset images are representative kidney sections stained for *Smad7* by ISH (RNAscope) n=3/3 for control/ΔMdm2^Hep respectively, dashed lines highlight renal tubules. **(c)** Representative images of dual p21 IHC/*Smad7* ISH in the kidney; pink arrows highlight examples of *Smad7* detection in red, whilst p21 is detected by brown 3,3′-Diaminobenzidine (DAB) staining; n=3/3 for control/ΔMdm2^Hep respectively. **(d)** UMAP plots showing the distribution of the cells that have a positive score for the JAK-STAT gene signature in the control (left) and the ΔMdm2^Hep model (right) renal cells. Inset images are representative kidney sections stained for pSTAT3 by IHC; n=3/3 for control/ΔMdm2^Hep respectively. Scale bars are 50μm and 5μm in inset magnification. **(e)** Automated quantification of pSTAT3⁺ in renal cortical cells; data are presented as percentage of total cortical cells, n=6 and 7 in control and ΔMdm2^Hep mice respectively; two-tailed Mann-Whitney test. Bars are mean ± S.E.M. and the numbers on the graphs are p values. **(f)** GSEA plot for a TGFβ

signalling pathway gene set on the significant differentially expressed genes from the bulk liver RNA-seq in the ΔMdm2^Hep model versus control at day 4. **(g)** Western blots for pSMAD2 and pSMAD3 and their respective total protein on whole kidney lysates from ΔMdm2^Hep model and control mice at day 4. n=5 mice in each group; see experimental schematic in Fig. 1a. 1 gel was run for pSMAD2/SMAD2 and another one for pSMAD3/SMAD3. β-actin was used as a loading control on both gels. **(h)** Representative images of ISH (RNAScope) for *Tgfb1* and *Tgfb2* on liver sections of ΔMdm2^Hep and control mice detected by DAB (brown); n=3/3 for control/ΔMdm2^Hep respectively for each ISH. **(i)** Representative images of dual p21 IHC/*Tgfb1* ISH (RNAScope) in the liver; *Tgfb1* ligand detection in red, whilst p21 is detected by brown DAB staining; n=3/3 for control/ΔMdm2^Hep respectively. Normalised *Tgfbr1*, *Tgfbr2* and *Tgfbr3* transcript counts (FPKMs) from bulk liver **(j)** and kidney **(k)** at day 4 in the ΔMdm2^Hep model versus control, n=4 except n=3 in liver control; both unpaired two-tailed t-test. All bars are mean ± S.E.M and the numbers on the graphs are p values. **(l)** Representative images of ISH (RNAScope) for *Tgfbr1* on kidney sections of ΔMdm2^Hep model and control mice at day 4 detected by brown DAB staining; n=3/3 for control/ΔMdm2^Hep respectively. Dashed lines highlight renal tubules. Scale bars are 50μm. Source numerical data and unprocessed blots are available in source data.

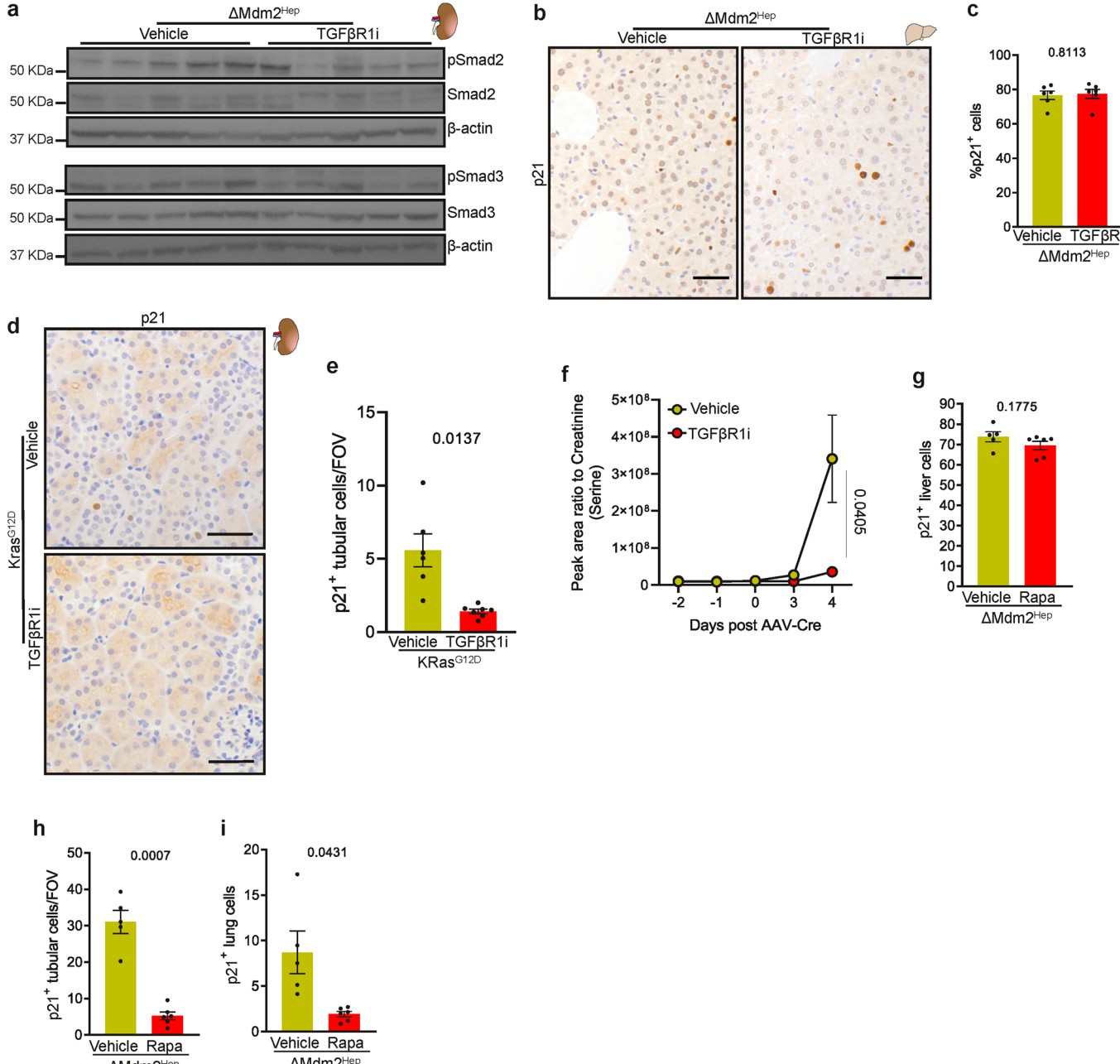

**Extended Data Fig. 8 | Systemic inhibition of the TGFβ signalling pathway does not affect liver senescence; see experimental schematic in Fig. 5a.**
(**a**) Western blots for pSMAD2, pSMAD3 and their non-phosphorylated forms on whole kidney lysates from vehicle- and TGFβR1i-treated ΔMdm2[Hep] mice. n=5 mice in each group. 1 gel was run for pSMAD2/SMAD2 and another one for pSMAD3/SMAD3. β-actin was used as a loading control on both gels. (**b**) Representative images of p21 IHC on liver sections of vehicle- and TGFβR1i-treated ΔMdm2[Hep] mice. (**c**) Automated quantification of p21[+] liver cells on liver sections of vehicle- and TGFβR1i-treated ΔMdm2[Hep] mice: data are presented as percentage of total liver cells, n=6 mice for each group; unpaired two-tailed t-test. (**d**) Representative images of p21 IHC on kidney sections of vehicle- and TGFβR1i-treated Kras[G12D] mice. (**e**) Manual quantification of p21[+] renal tubular cells in vehicle- and TGFβR1i-treated Kras[G12D] mice: data are presented as mean p21[+] tubular cells per 20 FOVs, n=6 and 7 vehicle- and TGFβR1i-treated Kras[G12D] mice respectively; two-tailed Welch's t-test. (**f**) Urine levels of Serine in ΔMdm2[Hep]

model over time, before and after AAV-Cre injection (Day 0). n=4 and 5 vehicle- and TGFβR1i-treated mice, respectively, per time point; two-tailed Welch's t-test comparing vehicle to TGFβR1i-treated mice at day 4. (**g**) Automated quantification of p21[+] liver cells on liver sections of vehicle and rapamycin-treated ΔMdm2[Hep] model mice 4 days after AAV induction (2x10[11] GC/mouse); data are presented as percentage of total liver cells, n=5 and n=6 vehicle- and rapamycin-treated mice respectively; two-tailed Mann-Whitney test. (**h**) Manual quantification of p21[+] renal cortical cells on kidney sections of vehicle- and rapamycin-treated ΔMdm2[Hep] mice: n=5 and n=6 vehicle- and rapamycin-treated mice respectively; two-tailed Welch's t-test. (**i**) Automated quantification of p21[+] lung cells on lung sections of vehicle- and rapamycin-treated ΔMdm2[Hep] mice: data are presented as percentage of total lung cells, n=5 and n=6 vehicle- and rapamycin-treated mice respectively; two-tailed Welch's t-test. Throughout all bars are mean ± S.E.M. and the numbers on the graphs are p values. Scale bars are 50μm. Source numerical data and unprocessed blots are available in source data.

| | |
|---|---|

# Reporting Summary

## Statistics

For all statistical analyses, confirm that the following items are present in the figure legend, table legend, main text, or Methods section.

| n/a | Confirmed | |
|---|---|---|
| ☐ | ☒ | The exact sample size (*n*) for each experimental group/condition, given as a discrete number and unit of measurement |
| ☐ | ☒ | A statement on whether measurements were taken from distinct samples or whether the same sample was measured repeatedly |
| ☐ | ☒ | The statistical test(s) used AND whether they are one- or two-sided<br>*Only common tests should be described solely by name; describe more complex techniques in the Methods section.* |
| ☐ | ☒ | A description of all covariates tested |
| ☐ | ☒ | A description of any assumptions or corrections, such as tests of normality and adjustment for multiple comparisons |
| ☐ | ☒ | A full description of the statistical parameters including central tendency (e.g. means) or other basic estimates (e.g. regression coefficient) AND variation (e.g. standard deviation) or associated estimates of uncertainty (e.g. confidence intervals) |
| ☐ | ☒ | For null hypothesis testing, the test statistic (e.g. *F*, *t*, *r*) with confidence intervals, effect sizes, degrees of freedom and *P* value noted<br>*Give P values as exact values whenever suitable.* |
| ☒ | ☐ | For Bayesian analysis, information on the choice of priors and Markov chain Monte Carlo settings |
| ☒ | ☐ | For hierarchical and complex designs, identification of the appropriate level for tests and full reporting of outcomes |
| ☒ | ☐ | Estimates of effect sizes (e.g. Cohen's *d*, Pearson's *r*), indicating how they were calculated |

*Our web collection on statistics for biologists contains articles on many of the points above.*

## Software and code

Policy information about availability of computer code

| Data collection | Open source code used with R environment |
|---|---|
| Data analysis | CellRanger (v.4.0), DropletUtils (v.1.10), Scater package (v.1.18) and the R environment (v.4.0) were used to analyse the scRNA-seq data. For the bulk RNA-seq, the R environment (v.3.4) was used to analyse the data, utilizing packages from Bioconductor (vXXX) and Metacore (https://portal.genego.com/). Raw data quality checks/trimming used FastQC (v0.11.7), FastP and FastQ Screen (v0.12.0). The reads were aligned to the mouse genome and annotation (vGRCm38.92) using HiSat2 (v2.1.0183).Determination and statistical analysis of expression levels was done by a combination of HTSeq (v0.9.1184), the R environment version (v3.4), utilizing packages from Bioconductor and differential gene expression analysis based on the negative binomial distribution using the DESeq2 package (v1.18.1186). Pathway Analysis was preformed using MetaCore (v) from Clarivate Analytics (https://portal.genego.com/). |

For manuscripts utilizing custom algorithms or software that are central to the research but not yet described in published literature, software must be made available to editors and reviewers. We strongly encourage code deposition in a community repository (e.g. GitHub). See the Nature Portfolio guidelines for submitting code & software for further information.

...

...

## Data

...

Policy information about availability of data

All manuscripts must include a data availability statement. This statement should provide the following information, where applicable:

- Accession codes, unique identifiers, or web links for publicly available datasets
- A description of any restrictions on data availability
- For clinical datasets or third party data, please ensure that the statement adheres to our policy

The fastq files and processed data for the single-cell RNAseq analysis of mouse kidney cells and bulk tissue transcriptomics in liver and kidney can be found on the Gene Expression Omnibus (GEO) repository; accession numbers: GSE189726, GSE267196 and GSE262705. Source data have been provided in Source Data. All other data generated and/or analysed during the current study are available from the corresponding author on reasonable request.

## Research involving human participants, their data, or biological material

Policy information about studies with human participants or human data. See also policy information about sex, gender (identity/presentation), and sexual orientation and race, ethnicity and racism.

| | |
|---|---|
| Reporting on sex and gender | Here we report the gender of the 34 patients enrolled in the study as defined by their self identification as recorded in medical records. |
| Reporting on race, ethnicity, or other socially relevant groupings | No socially constructed or socially relevant categorisations are made. |
| Population characteristics | atient covariates are described in table 1. Patients were recruited between 2010 and 2019. The diagnosis of acute indeterminate hepatitis was made by a combination of clinical, biochemical and histopathological criteria. The genders of the patients were balanced, with 7/10 male/females in each group. |
| Recruitment | The study included 34 consecutive patients with severe acute indeterminate hepatitis who were admitted to a single hospital and underwent transjugular liver biopsy or liver transplantation. |
| Ethics oversight | The study was approved by the London-Hampstead Research Ethics Committee (07/Q0501/50) and was in accordance with the declaration of Helsinki. |

Note that full information on the approval of the study protocol must also be provided in the manuscript.

# Field-specific reporting

Please select the one below that is the best fit for your research. If you are not sure, read the appropriate sections before making your selection.

☒ Life sciences ☐ Behavioural & social sciences ☐ Ecological, evolutionary & environmental sciences

For a reference copy of the document with all sections, see nature.com/documents/nr-reporting-summary-flat.pdf

# Life sciences study design

All studies must disclose on these points even when the disclosure is negative.

| | |
|---|---|
| Sample size | For animal experiments power calculations were not routinely performed; however, throughout animal biological replicate numbers were chosen based upon a predicted magnitude of response (50% effect size typically) taking into account the variability observed in pilot and prior experiments using control mice. This included analysis of mice induced outside the 8-12 week age induction window (additional data from these mice is available upon request). For all experiments the number of biological replicates ≥ 3 mice per cohort, specific n numbers for biological replicates are reported in figure legends for each experiment. |
| Data exclusions | Aside from the quality control steps using the pre-established exclusion criteria described no data was excluded from the analyses. |
| Replication | In animal experiments the N number reflects biological replicates, mice came from different litters and thus were not always sampled/induced on the same day. Staining and analysis was performed for all IHC analyses. Consistency was observed between between separate experiments when staining and analyses were batched. Combined data is reported throughout. Data from mice induced outside the 8-12 week age induction window was also analysed but not included here. This was typically consistent with those reported here also (additional data for these available upon request). All repeated attempts at induction over time were successful and no animals were excluded from analyses. For qRT-PCR data, N reflects biological replicates (mice) but within each plate, three technical replicates were performed for each biological replicate. For the serum treatment of wt MEFs (Fig. 4b), each data point in the "control" and "ΔMdm2Hep" groups represent technical replicates which derive from 2 different biological replicates i.e. plasma samples (in each group, there are 2 technical replicates for each one of the 2 biological replicates; this is highlighted specifically in the legend). In the "no plasma" group in Fig. 3b each dot represents a technical replicate (all technical replicates for this group were performed on the same plate). For the plasma treatment of the neural stem cell derived neuronal cells (Fig. 4c), each dot represents the mean of 3 technical replicates from one biological replicate i.e. plasma samples. In the "no |

plasma" and "H2O2" groups in Fig. 4c, each dot represents a technical replicate (all technical replicates for these groups were performed on the same plate).

| | |
|---|---|
| Randomization | Mice were manually split between experimental groups within litters and group sizes balanced overall. Biases were controlled as mice were assigned experimental group based on mouse ID which was assigned randomly by independent animal unit staff. Single colonies of inbred animals were used throughout, thus minimising bias between litters. Batches of AAV were created from common quantified stock solutions. All cohorts used in this study were age 8-12 weeks old unless otherwise stated in methods and male, except for the mice of the KrasG12D mouse model which were both male and female. |
| Blinding | Throughout animal welfare checks experimenters we not blinded to experimental groups. Group allocation is described above. Technical staff administering therapy were blinded to experimental conditions of mice. The experimenters were not blinded as induced mice with the LSL-RFP reporter develop red skin at the time of sampling. All subsequent tissue handling and analysis was blinded and/or performed using standardised automated analyses where possible. |

# Reporting for specific materials, systems and methods

We require information from authors about some types of materials, experimental systems and methods used in many studies. Here, indicate whether each material, system or method listed is relevant to your study. If you are not sure if a list item applies to your research, read the appropriate section before selecting a response.

## Materials & experimental systems

| n/a | Involved in the study |
|---|---|
| ☐ | ☒ Antibodies |
| ☐ | ☒ Eukaryotic cell lines |
| ☒ | ☐ Palaeontology and archaeology |
| ☐ | ☒ Animals and other organisms |
| ☐ | ☒ Clinical data |
| ☒ | ☐ Dual use research of concern |
| ☒ | ☐ Plants |

## Methods

| n/a | Involved in the study |
|---|---|
| ☒ | ☐ ChIP-seq |
| ☒ | ☐ Flow cytometry |
| ☒ | ☐ MRI-based neuroimaging |

## Antibodies

| | |
|---|---|
| Antibodies used | Antibody details are also provided in Table S3.<br>Antibody (primary)/ Supplier /Catalogue No /Dilution /Application /Clone number (monoclonals)<br>BrdU BD Biosciences 347580 1:250 IHC/IF B44<br>Caspase 3 Cell Signaling 9661 1:500 IHC NA<br>p21 Abcam ab107099 1:150 IHC/IF HUGO291<br>p53 Leica NCL-L-p53CM5p 1:750 IHC NA<br>pSMAD2 Cell Signalling 3108S 1:1000 WB 138D4<br>pSMAD3 Abcam ab52903 1:1000 WB EP823Y<br>pSTAT3 Cell Signaling 9131 1:100 IHC NA<br>RFP Tebu-Bio 600-401-379 1:1000 IHC/IF NA<br>SMAD2 Cell Signalling 5339 1:1000 WB D43B4<br>SMAD3 Cell Signalling 9513S 1:1000 WB NA<br>β-actin Sigma-Aldrich A2228 1:2000 WB AC-74<br>Lrp2 Abcam ab76969 1:1000 IF NA<br>Calb1 Abcam ab229915 1:1000 IF EPR22698-236<br><br>Antibody (secondary) Supplier Catalogue No Dilution Application<br>Donkey anti-mouse 488 Life Technologies A-21202 1:200 IHC<br>Donkey anti-mouse 555 Life Technologies A-31570 1:200 IHC<br>Donkey anti-rabbit 647 Life Technologies A-31573 1:200 IHC<br>Goat anti mouse HRP Cell Signalling 7076 1:3000 WB<br>Goat anti rabbit HRP Cell Signalling 7074 1:3000 WB<br>Horse anti-rabbit biotinylated Vector Labs BP-1100 R.T.U IHC |
| Validation | RFP; 600-401-379 – Manufacturer website: Application suitable for: IF and IHC. Reacts with: Mouse. Validated In house on RFP negative mouse liver tissue<br>BrdU; 347580 - relevant publication using antibody for IHC in mouse tissue from FFPE sections, validated in house in mice that did not receive BrdU cell labelling agent (negative control)<br>Cleaved caspase 3; 9661 - Manufacturer website: Applications: WB, IP, IHC-P, IF-IC, F. Reacts with: human, Mouse, Rat, Monkey. Mouse monoclonal antibody.<br>Lipocalin-2; AF1857 SP - Manufacturer website: Detects mouse Lipocalin-2/NGAL in direct ELISAs and Western blots. Goat polyclonal antibody, reacts with mouse.<br>p21; ab107099 - Manufacturer website: Suitable for: WB, IHC-P. Rat monoclonal, reacts with mouse and human. Validated in house on kidney tissue (IHC) from a p21 KO mouse and on whole liver lysate (WB) of another p21 KO mouse.<br>p53; NCL-L-p53CM5p - Manufacturer website: Suitable for IHC-P (HIER), rabbit polyclonal, reacts with mouse and rat.<br>pSMAD2; 3108S - Manufacturer website: Suitable for WB. Reacts with human, mouse, rat and mink. Rabbit monoclonal. |

pSMAD3; ab52903 - Manufacturers website: Suitable for: WB, ICC/IF, IHC-P, Dot blot. Reacts with: Mouse, Human. Rabbit monoclonal.
pSTAT3; 9131 - Manufacturer website: Suitable for: WB, IP, ChIP. Reacts with human, mouse, rat and mink. Rabbit monoclonal.
SMAD2; 5339 - Manufacturer website: Suitable for: WB, IP, IF, ChIP, Flow cytometry. Reacts with human, mouse, rat and mink. Rabbit monoclonal.
SMAD3; 9513S - Manufacturer website: Suitable for: WB, IP, IF. Reacts with human, mouse and rat. Rabbit monoclonal.
β-actin; A2228 - Manufacturer website: Application suitable for: as loading control for western blots. It is a mouse monoclonal antibody.
Lrp2; Ab76969 - Manufacturer website: Suitable for: IHC-P. Reacts with: Mouse, Rat, Human, Monkey. Rabbit polyclonal.
Calb1; Ab 229915 - Manufactuer website: Suitable for: IHC-Fr, IHC-P, WB, IP. Reacts with: Mouse, Rat. Rabbit monoclonal

# Eukaryotic cell lines

Policy information about cell lines and Sex and Gender in Research

| Cell line source(s) | The WT murine embryonic fibroblasts (MEFs) were derived from E13.5-14.5 wt C57Bl/6J mouse embryos (gender not specified) and the neuronal stem cell - derived neuronal cells were derived from male human fetal neural crest progenitor cells as referenced. |
|---|---|
| Authentication | None of the cell lines were authenticated |
| Mycoplasma contamination | The wt MEFs were confirmed to be free of mycoplasma contamination at passage 4 by a PCR-based assay. The iPSCs-derived neuronal cells were isolated and differentiated as described in (Madgwick A. at al., 2015) and were not tested for mycoplasma contamination. |
| Commonly misidentified lines (See ICLAC register) | No commonly misidentifed cell lines were used in this study |

# Animals and other research organisms

Policy information about studies involving animals; ARRIVE guidelines recommended for reporting animal research, and Sex and Gender in Research

| Laboratory animals | The animals used in this study were on a mixed (129P2/OlaHsdWtsi;C57Bl/6J) background. Only male animals were used except for the experiments with the KrasG12D mouse model were both males and females were used. All mice used in this study were induced between 8-12 weeks of age. The mice used in this study carried the following transgenes: Mdm2tm2.1Glo, Gt(ROSA)26Sortm14(CAG-tdTomato)Hze and Krastm4Ty. All transgenic mice used were born in house. |
|---|---|
| Wild animals | This study did not involve wild animals. |
| Reporting on sex | Sex of animals is reported throughout. As outlined above only male animals were used except for the experiments with the KrasG12D mouse model were both males and females were used. |
| Field-collected samples | This study did not involve samples collected from the field. |
| Ethics oversight | All animal studies were performed in accordance with a UK Home Office project licence (70/8891 (protocol number 2) or PP0604995 (protocol number 3)) and were subject to review by the animal welfare and ethical review board of the University of Glasgow. |

Note that full information on the approval of the study protocol must also be provided in the manuscript.

# Clinical data

Policy information about clinical studies
All manuscripts should comply with the ICMJE guidelines for publication of clinical research and a completed CONSORT checklist must be included with all submissions.

| Clinical trial registration | Details of a clinical study (non CTIMP) are reported. Clinical Trial Registration: the ethics pre-dates requirement for clinical trials registration (2007). |
|---|---|
| Study protocol | Entitled "Biomarkers of hepatic inflammation, senescence, and regeneration in severe acute hepatitis" is available upon request from Professor Jalan., Royal Free Hospital London |
| Data collection | Data collection was performed from patient notes both prospectively and retrospectively. Data collection form is available upon request from Professor Jalan, Royal Free Hospital London. |
| Outcomes | The study is observational upto 90 days post admission to hospital and patients continue to be followed up without prespecified endpoints to 90 days or until mortality. |

## Plants

Seed stocks
> *Report on the source of all seed stocks or other plant material used. If applicable, state the seed stock centre and catalogue number. If plant specimens were collected from the field, describe the collection location, date and sampling procedures.*

Novel plant genotypes
> *Describe the methods by which all novel plant genotypes were produced. This includes those generated by transgenic approaches, gene editing, chemical/radiation-based mutagenesis and hybridization. For transgenic lines, describe the transformation method, the number of independent lines analyzed and the generation upon which experiments were performed. For gene-edited lines, describe the editor used, the endogenous sequence targeted for editing, the targeting guide RNA sequence (if applicable) and how the editor was applied.*

Authentication
> *Describe any authentication procedures for each seed stock used or novel genotype generated. Describe any experiments used to assess the effect of a mutation and, where applicable, how potential secondary effects (e.g. second site T-DNA insertions, mosiacism, off-target gene editing) were examined.*

