## [Peer Review File · Nature Cell Biology]

Peer Review Information

Journal: Nature Cell Biology

Manuscript Title: Hepatocellular senescence induces multi-organ senescence and dysfunction via TGF β

Corresponding author name(s): Professor Thomas Bird

Editorial Notes:

Redactions – transferred manuscripts (mention of previous referee reports from elsewhere) This manuscript has been previously reviewed at another journal. This document only contains reviewer comments, rebuttal and decision letters for versions considered at Nature Cell Biology. Mentions of prior referee reports have been redacted

Redactions – published data Parts of this Peer Review File have been redacted as indicated to remove third-party material.

Reviewer Comments & Decisions:

Decision Letter, initial version:

Our ref: NCB-A54893-T

8th August 2024

Dear Dr. Bird,

Thank you again for submitting your revised manuscript "Inter-organ transmission of hepatocellular senescence induces multi-organ dysfunction through the TGF β signalling pathway." (NCB-A54893-T) to Nature Cell Biology. As you know, unfortunately, the reviewers from [REDACTED] were not available to re-review. I am sorry about this. We recruited a new expert, Rev#5 (senescence, metabolism), to help us assess the changes to all previous reviewer comments from the latest round of review at [REDACTED].

The revision has now been seen by that expert and their comments are below. The reviewer finds that the paper has improved in revision: you will see they find the revisions adequate but point out caveats and confounding factors. It will be important to incorporate the pertinent caveats about data interpretation into the discussion and acknowledge the need for further research to address these questions. Please also follow-up on the suggested analysis of the proportion of cells positive for p21 that are positive for another marker of senescence and TGFbeta and what proportion of cells that are

positive for another marker of senescence are p21+, using existing datasets.

Overall, we'll be happy in principle to publish the manuscript in Nature Cell Biology pending minor revisions to satisfy the referee's final requests as explained above and to comply with our editorial and formatting guidelines. With your final resubmission (after we send you our checklist and you make necessary formatting changes), please include a point-by-point response to the reviewer's comments and indicate clearly in this rebuttal what changes were made to the manuscript and where these changes were made in the manuscript to fast-track our editorial assessment of the changes.

We are now performing detailed checks on your paper and will send you a checklist detailing our editorial and formatting requirements in about 1-2 weeks. Please do not upload the final materials and make any revisions until you receive this additional information from us.

Thank you again for your interest in Nature Cell Biology. Please do not hesitate to contact me if you have any questions.

Sincerely,

Melina

Melina Casadio, PhD
Senior Editor, Nature Cell Biology
ORCID ID: <https://orcid.org/0000-0003-2389-2243>

Reviewer #5 (Remarks to the Author):

This body of work is provocative and has important translational relevance. The authors have done a good job responding to the previous critique. The data in the revised manuscript clearly show that selective deletion of mdm2 in hepatocytes reduces kidney and brain function, induces p21 in subpopulations of hepatocytes and proximal tubular cells (PTCs) in the kidney, and somewhat increases beta gal staining in the liver and kidney. Changes in the level of mdm2 suppression in the liver also modulate the aforementioned parameters in the extrahepatic organs. Although TGFb is clearly involved in the multi-organ failure (MOF) that occurs when mdm2 is deleted in hepatocytes (or when Kras is over-expressed in hepatocytes to promote senescence), it is less clear to me that the liver-driven MOF is due to hepatocyte senescence (or hepatocyte-derived TGFb), per se.

The authors focus on p21(+) hepatocytes as drivers of MOF and suggest that this occurs because deleting mdm2 stabilizes p53 which, in turn, upregulates p21 and the latter induces hepatocyte senescence. However, they also report that deleting p21 does not reverse the effects of hepatocyte mdm2 deletion on the renal (or hepatic) phenotypes. This is an important finding as it suggests that some other target of mdm2 may be responsible and as a ubiquitin ligase, mdm2 is known to regulate the stability of multiple proteins. Similarly, although Kras overexpression increases p21, it also dysregulates the expression of many other genes.

Data interpretation is further confounded by the fact that p21 induction does not always cause cell senescence (i.e., irreversible cell cycle arrest). Rather, p21 induction may simply slow cell cycle progression transiently to 'buy time' for cellular reprogramming. The latter often involves partial epithelial-mesenchymal state transitions that are accompanied by changes in growth inhibitory/pro-fibrogenic factors, including tgfb. The secretome of the growth-inhibited/transitioning cells can evoke state transitions (including senescence) in neighboring cells. Thus, it seems important to determine: i) what % of the p21(+) hepatocytes (or PTCs) are B-gal(+) and TGFb(+) and conversely, what % of the

b-gal(+) cells are p21(+) - this could be done using immunohistochemistry. These questions can also be addressed using the single cell RNA seq data (e.g., what % of the p21(+) hepatocyte subpopulation express other markers of senescence and TGF β (+). This issue merits consideration, as the manuscript reports that subpopulations of PTCs that are p21(+) are not the PTC subpopulation that is most enriched with another, well-validated score for PTC senescence.

Decision Letter, first revision:

Our ref: NCB-A54893-T

13th August 2024

Dear Dr. Bird,

Thank you for your patience as we've prepared the guidelines for final submission of your Nature Cell Biology manuscript, "Inter-organ transmission of hepatocellular senescence induces multi-organ dysfunction through the TGF β signalling pathway." (NCB-A54893-T). Please carefully follow the step-by-step instructions provided in the attached file, and add a response in each row of the table to indicate the changes that you have made. Ensuring that each point is addressed will help to ensure that your revised manuscript can be swiftly handed over to our production team.

In recognition of the time and expertise our reviewers provide to Nature Cell Biology's editorial process, we would like to formally acknowledge their contribution to the external peer review of your manuscript entitled "Inter-organ transmission of hepatocellular senescence induces multi-organ dysfunction through the TGF β signalling pathway.". For those reviewers who give their assent, we will be publishing their names alongside the published article.

Nature Cell Biology offers a Transparent Peer Review option for new original research manuscripts submitted after December 1st, 2019. As part of this initiative, we encourage our authors to support increased transparency into the peer review process by agreeing to have the reviewer comments, author rebuttal letters, and editorial decision letters published as a Supplementary item. When you submit your final files please clearly state in your cover letter whether or not you would like to participate in this initiative. Please note that failure to state your preference will result in delays in accepting your manuscript for publication.

Cover suggestions

COVER ARTWORK: We welcome submissions of artwork for consideration for our cover. For more information, please see our guide for cover artwork.

Nature Cell Biology has now transitioned to a unified Rights Collection system which will allow our Author Services team to quickly and easily collect the rights and permissions required to publish your work. Approximately 10 days after your paper is formally accepted, you will receive an email in providing you with a link to complete the grant of rights. If your paper is eligible for Open Access, our Author Services team will also be in touch regarding any additional information that may be required to arrange payment for your article.

Please note that *Nature Cell Biology* is a Transformative Journal (TJ). Authors may publish their research with us through the traditional subscription access route or make their paper immediately open access through payment of an article-processing charge (APC). Authors will not be required to make a final decision about access to their article until it has been accepted. Find out more about Transformative Journals

Please use the following link for uploading these materials:
[Redacted]

Best regards,

Kendra Donahue
Staff
Nature Cell Biology

On behalf of

Melina Casadio, PhD
Senior Editor, Nature Cell Biology
ORCID ID: <https://orcid.org/0000-0003-2389-2243>

Reviewer #5:

Remarks to the Author:

This body of work is provocative and has important translational relevance. The authors have done a good job responding to the previous critique. The data in the revised manuscript clearly show that selective deletion of mdm2 in hepatocytes reduces kidney and brain function, induces p21 in subpopulations of hepatocytes and proximal tubular cells (PTCs) in the kidney, and somewhat increases beta gal staining in the liver and kidney. Changes in the level of mdm2 suppression in the liver also modulate the aforementioned parameters in the extrahepatic organs. Although TGFb is clearly involved in the multi-organ failure (MOF) that occurs when mdm2 is deleted in hepatocytes (or when Kras is over-expressed in hepatocytes to promote senescence), it is less clear to me that the liver-driven MOF is due to hepatocyte senescence (or hepatocyte-derived TGFb), per se.

The authors focus on p21(+) hepatocytes as drivers of MOF and suggest that this occurs because

deleting mdm2 stabilizes p53 which, in turn, upregulates p21 and the latter induces hepatocyte senescence. However, they also report that deleting p21 does not reverse the effects of hepatocyte mdm2 deletion on the renal (or hepatic) phenotypes. This is an important finding as it suggests that some other target of mdm2 may be responsible and as a ubiquitin ligase, mdm2 is known to regulate the stability of multiple proteins. Similarly, although Kras overexpression increases p21, it also dysregulates the expression of many other genes.

Data interpretation is further confounded by the fact that p21 induction does not always cause cell senescence (i.e., irreversible cell cycle arrest). Rather, p21 induction may simply slow cell cycle progression transiently to 'buy time' for cellular reprogramming. The latter often involves partial epithelial-mesenchymal state transitions that are accompanied by changes in growth inhibitory/pro-fibrogenic factors, including tgfb. The secretome of the growth-inhibited/transitioning cells can evoke state transitions (including senescence) in neighboring cells. Thus, it seems important to determine: i) what % of the p21(+) hepatocytes (or PTCs) are B-gal(+) and TGFb(+) and conversely, what % of the b-gal(+) cells are p21(+) - this could be done using immunohistochemistry. These questions can also be addressed using the single cell RNA seq data (e.g., what % of the p21(+) hepatocyte subpopulation express other markers of senescence and TGFb(+)). This issue merits consideration, as the manuscript reports that subpopulations of PTCs that are p21(+) are not the PTC subpopulation that is most enriched with another, well-validated score for PTC senescence.

Author Rebuttal to Initial comments

Editors comments

It will be important to incorporate the pertinent caveats about data interpretation into the discussion and acknowledge the need for further research to address these questions. Please also follow-up on the suggested analysis of the proportion of cells positive for p21 that are positive for another marker of senescence and TGFbeta and what proportion of cells that are positive for another marker of senescence are p21+, using existing datasets.

With your final resubmission (after we send you our checklist and you make necessary formatting changes), please include a point-by-point response to the reviewer's comments and indicate clearly in this rebuttal what changes were made to the manuscript and where these changes were made in the manuscript to fast-track our editorial assessment of the changes.

Reviewers comments

This body of work is provocative and has important translational relevance. The authors have done a good job responding to the previous critique. The data in the revised manuscript clearly show that selective deletion of mdm2 in hepatocytes reduces kidney and brain function, induces p21 in subpopulations of hepatocytes and proximal tubular cells (PTCs) in the kidney, and somewhat increases beta gal staining in the liver and kidney. Changes in the level of mdm2 suppression in the liver also modulate the aforementioned parameters in the extrahepatic organs. Although TGFb is clearly involved in the multi-organ failure (MOF) that occurs when mdm2 is deleted in hepatocytes (or when Kras is over-expressed in hepatocytes to promote senescence), it is less clear to me that the liver-driven MOF is due to hepatocyte senescence (or hepatocyte-derived TGFb), per se.

The authors focus on p21(+) hepatocytes as drivers of MOF and suggest that this occurs because deleting mdm2 stabilizes p53 which, in turn, upregulates p21 and the latter induces hepatocyte senescence. However, they also report that deleting p21 does not reverse the effects of hepatocyte mdm2 deletion on the renal (or hepatic) phenotypes. This is an important finding as it suggests that some other target of mdm2 may be responsible and as a ubiquitin ligase, mdm2 is known to regulate the stability of multiple proteins. Similarly, although Kras overexpression increases p21, it also dysregulates the expression of many other genes.

Data interpretation is further confounded by the fact that p21 induction does not always cause cell senescence (i.e., irreversible cell cycle arrest). Rather, p21 induction may simply slow cell cycle progression transiently to 'buy time' for cellular reprogramming. The latter often involves partial epithelial-mesenchymal state transitions that are accompanied by changes in growth inhibitory/pro-fibrogenic factors, including tgfb. The secretome of the growth-inhibited/transitioning cells can evoke state transitions (including senescence) in neighboring cells. Thus, it seems important to determine: i) what % of the p21(+) hepatocytes (or PTCs) are B-gal(+) and TGFb(+) and conversely, what % of the b-gal(+) cells are p21(+) - this could be done using immunohistochemistry. These questions can also be addressed using the single cell RNA seq data (e.g., what % of the p21(+) hepatocyte subpopulation express other markers of senescence and TGFb(+)). This issue merits consideration, as the manuscript reports that

subpopulations of PTCs that are p21(+) are not the PTC subpopulation that is most enriched with another, well-validated score for PTC senescence.

Response to reviewer's comments:

We are grateful for the comments above and pleased to provide further detail to address this directly. With respect to the senescence phenotype in the liver we have shown that this is widespread using a variety of markers but have not to date done the direct comparison between the senescence markers suggested by the reviewer. In our previous work we show that Mdm2 deletion in the liver is associated with p21 expression globally in hepatocytes and this is associated with similar levels of other senescence markers including γ H2Ax, IL1 α and SA- β gal (Rebuttal Figure 1).

[REDACTED]

We have now quantified the coexpression of γ H2Ax and p21 in the hepatocyte population in the Mdm2 model used in the current manuscript (rebuttal Figure 2). We have elected this approach as the expression of γ H2Ax is an accepted senescence marker and facilitates accurate individual cell co-registration unlike SA- β gal. It has the added advantage of colocalization of these nuclear markers overcoming the problems of cell segmentation and registration. We highlight however that, as shown above, the results are likely to be highly similar whichever of these senescence

markers is used because they are almost universally expressed by all hepatocytes at day 4 in this model (panels b and c above).

From our new and previous analyses, we find that there is indeed very high correlation between γ H2Ax, SA- β gal and p21 senescence markers in the liver in this model. Our updated analysis shows this directly by the two markers we can accurately test using dual IHC; this finds that there are minor populations with either γ H2Ax or p21 alone, consistent with previous data^{1,2}. We have not added these new data to the revision for two reasons, firstly because it is essentially shown in the previous publication described above¹. Secondly because the induction of each senescence marker is essentially universal in the hepatocyte population their co-presence in individual hepatocytes is implicit and predictable in the liver in this model.

Rebuttal Figure 2: Assessment of dual senescence markers in liver. Left panel. Quantification of multiple senescence markers (γ H2Ax and p21) identified by immunohistochemistry in hepatocytes 4 days following AAV-TBG-Cre induction (2×10^{11} GC/mouse) in the Mdm2 model. Liver sections were co-stained for γ H2Ax (red) and p21 (brown). Right panel. Images were quantified following colour deconvolution using ImageJ and cell co-registered. Bars are mean \pm S.E.M., each dot represents one biological replicate (mean of 2×20 fields, approx. 509-614 cells per replicate). Representative image of staining is shown. Arrow highlights example of a γ H2Ax⁺/p21⁻ hepatocyte. Scale bar = 50 μ m.

There are previous single cell transcriptomic studies (e.g. Teo *et al.*³) which provide some insights into the transcriptomic state of the cells in the model but as these models were performed in lower levels of hepatocellular Mdm2 deletion they do not represent suitable data sets to address this question from a transcriptomic standpoint. We are pleased however, to provide further quantitative data as described above to directly address this specific question in the hepatocyte population.

With respect to TGF β status of hepatocytes, we believe this is fundamentally not the right approach for two reasons. Firstly, TGF β production in the model is predominantly from a non-hepatocellular population and secondly it is fundamentally more challenging to ascribe a positive versus negative status to this population as this is unlikely to be a binary phenomenon. Regarding the first point, our work and that of others has shown that in the healthy liver (murine or human) there is a minority of hepatocytes which produce low levels of TGF β ligands (TGFB1/2/3

or Tgfb1/2/3) whilst the bulk of physiological TGF β ligands are produced by the non-parenchymal cells (The Liver Regeneration Atlas)^{4,5}. We show in our current manuscript that hepatic TGF β is increased in the liver as a whole (control/Mdm2 183/440; p=0.0005, Tgfb1 FPKMs from bulk hepatic transcriptome data for Tgfb1 for example – see source data file), as would be predicted from previous studies. This occurs both by senescent hepatocytes but also non-parenchymal cells; the latter being by far the majority site of production, principally monocytes; see previous rebuttals, Figure S7h (reproduced below in Rebuttal Figure 3) and also below in the APAP senescent model¹ and human disease¹ (Rebuttal Figure 4) and we state that this is a hepatic (but not a hepatocytic) process in the manuscript. There are hepatocytes in the Mdm2 model which also produce smaller amounts of Tgfb1 transcript, as we highlight in Figure S7i, but at no point do we claim that TGF β production is a hepatocyte-specific phenomenon. Given the model, however, this non-parenchymal production is clearly driven by a primary hepatocellular senescence process. We did highlight that our previous data also supported the primary site of TGF β being the non-parenchymal monocytic population in a previous rebuttal (point 1.5 in March 2024) but accept that this may not have been available to the current reviewer. Therefore, we feel that quantification of dual senescence marker and TGF β status of hepatocytes is not helpful.

Additionally, returning to our second concern about the arbitrary assignment of TGF β status, we also wish to highlight the challenge posed by quantifying from 2D sections positivity of TGF β ligands; this is the only realistic means we see of performing such an analysis, without resorting to further single cell sequencing. This method could produce an arbitrary TGF β -positive versus negative population using presence or absence of Tgfb1 in situ hybridisation (ISH) spots which could then be correlated to p21. However, even accounting for the problems of doing this analysis on 4 μ m thick '2D' sections (assessing the 30-50 μ m deep hepatocytes) for the presence/absence of only a very small number of hepatocyte Tgfb1 ISH spots (typically <1) with the associated senescence markers (accepting the additional issues of cell segmentation) does not allow us to confidently ascribe the 'TGF β status' to individual cells. We show examples where a relatively rare example of p21 hepatocytes clearly display multiple Tgfb1 ISH spots in Extended data Fig 7j, however the many other senescence hepatocytes will often not have any Tgfb1 ISH spots, but this does not mean that they are not producing TGF β for the reasons we describe above. Personally, I would feel very uncomfortable being asked to provide such quantitative information knowing that it is unlikely to provide an accurate reflection of the ground truth.

Rebuttal Figure 3: Images from Extended data figure 7h – ISH identification of Tgfb1 production in the liver at 4 days in the Mdm2 model versus control. ISH dots in brown localise with non-parenchymal cells predominantly and not hepatocytes. Scale bars denote 50 μ m.

[Redacted]

To separately address the question of senescence markers in the PTC population we have revisited our scRNAseq analyses and compared quantitatively the renal senescence signature and the p21 signatures as recommended by the reviewer (Rebuttal Figure 3). Within the PTC population and more generally, there is indeed poor overlap between these two populations in line with the reviewer's query as highlighted by these data and we have amended the figure legend to show these. We would highlight that the p21 signature does encompass of the smaller population of cells identified by the senescence signature within the PTC population.

Rebuttal Figure 3: Quantification of senescence associated signatures within the PTC population. Quantification of multiple senescence signatures in the renal populations 4 days following AAV-TBG-Cre induction (2×10^{11} GC/mouse) in the Mdm2 model. The p21 and renal senescence signatures are described in the main manuscript and relevant genes displayed in Extended Data Table S5 and in O'Sullivan, E. D. *et al.*¹⁴. Data are presented as means for the entire PTC population between 3 pooled biological replicates as described in the manuscript.

We have modified the discussion to highlight the caveats raised by the reviewer as outlined below.

Discussion

The SASP is a central mediator of the non-autonomous effects of senescent cells. Here we demonstrate that senescence can be transmitted to and affect the function of distant organs in a systemic manner. In the context of acute injury, senescence has often been described as part of a finely-tuned mechanism with overall beneficial effects for wound healing^{5,6}. SASP factors have been shown to induce reprogramming in neighbouring cells, facilitating tissue regeneration⁷⁻⁹. However, following severe injury, this mechanism may have the opposite effect systemically,

through excessive SASP production, including senescence itself. In turn, this excessive stimulus for senescence can be associated with compromised organ function.

Systemic transmission of senescence may be relevant to several diseases. Here we use a series of models of hepatocyte-specific senescence to model an acute senescence phenotype, such as the one observed during ALF. ALF is itself characterised by sequential multi-organ failure typically beginning with the kidney progressing to also involve the brain, lungs and other organs. This clinical progression may, at least in part, be underpinned by the systemic transmission of senescence. The data in patients with acute indeterminate hepatitis, showing that increased hepatocellular expression of p21 at initial presentation prior to multiorgan failure can predict ensuing multiorgan failure, requirement for liver transplant and/or death, provides evidence for a novel biomarker that both allows early risk stratification and selection of patients for specific therapies. Similarly, the observation that TGF β signalling is a central driver of systemic transmission of senescence may pave the way for new therapeutic approaches in diseases where this phenomenon occurs. ~~This~~ Whilst this effect may either be independent of solely p21-dependent senescence or a phenomenon related to TGF β activity outwith of senescence it is in line with the beneficial effects of senolytics and senomorphics that have been elegantly demonstrated on numerous pathologies¹⁰⁻¹³. Further research is required to dissect out the direct causal link between senescence in the primary tissue and the systemic effects and how they are affected by factors such as disease site or specific senescence phenotype, chronicity of senescence and the interaction with concurrent or pre-existing senescence in other organs. Our results demonstrate that systemic transmission of senescence can induce systemic organ dysfunction which may be central to multisystemic sequelae in many diseases.

- 1 Bird, T. G. *et al.* TGFbeta inhibition restores a regenerative response in acute liver injury by suppressing paracrine senescence. *Science translational medicine* **10**, doi:10.1126/scitranslmed.aan1230 (2018).
- 2 Lu, W. Y. *et al.* Hepatic progenitor cells of biliary origin with liver repopulation capacity. *Nat Cell Biol* **17**, 971-983, doi:10.1038/ncb3203 (2015).
- 3 Teo, Y. V. *et al.* Notch Signaling Mediates Secondary Senescence. *Cell Rep* **27**, 997-1007.e1005, doi:10.1016/j.celrep.2019.03.104 (2019).
- 4 Matchett, K. P. *et al.* Multimodal decoding of human liver regeneration. *Nature* **630**, 158-165, doi:10.1038/s41586-024-07376-2 (2024).
- 5 Remmerie, A. *et al.* Osteopontin Expression Identifies a Subset of Recruited Macrophages Distinct from Kupffer Cells in the Fatty Liver. *Immunity* **53**, 641-657.e614, doi:10.1016/j.immuni.2020.08.004 (2020).

Final Decision Letter:

Dear Dr Bird,

I am pleased to inform you that your manuscript, "Hepatocellular senescence induces multi-organ senescence and dysfunction via TGF β ", has now been accepted for publication in Nature Cell Biology.

Please note that *Nature Cell Biology* is a Transformative Journal (TJ). Authors may publish their research with us through the traditional subscription access route or make their paper immediately open access through payment of an article-processing charge (APC). Authors will not be required to make a final decision about access to their article until it has been accepted. Find out more about Transformative Journals

If you have not already done so, we strongly recommend that you upload the step-by-step protocols used in this manuscript to protocols.io (<https://protocols.io>), an open online resource that allows researchers to share their detailed experimental know-how. All uploaded protocols are made freely available and are assigned DOIs for ease of citation. Protocols and Nature Portfolio journal papers in which they are used can be linked to one another, and this link is clearly and prominently visible in the

online versions of both. Authors who performed the specific experiments can act as primary authors for the Protocol as they will be best placed to share the methodology details, but the Corresponding Author of the present research paper should be included as one of the authors. By uploading your Protocols onto protocols.io, you are enabling researchers to more readily reproduce or adapt the methodology you use, as well as increasing the visibility of your protocols and papers. You can also establish a dedicated workspace to collect your lab Protocols. Further information can be found at <https://www.protocols.io/help/publish-articles>.

With kind regards,

Melina Casadio, PhD
Senior Editor, Nature Cell Biology
ORCID ID: <https://orcid.org/0000-0003-2389-2243>
